# CD4 derived double negative T cells prevent the development and progression of nonalcoholic steatohepatitis

Guangyong Sun[1,2,3,4,5], Xinyan Zhao[6], Mingyang Li[2,3,4,5], Chunpan Zhang[2,3,4,5], Hua Jin[2,3,4,5], Changying Li[2,3,4,5], Liwei Liu[6], Yaning Wang[2,3,4,5], Wen Shi[2,3,4,5], Dan Tian [2,3,4,5], Hufeng Xu[2,3,4,5], Yue Tian[2,3,4,5], Yongle Wu[7], Kai Liu[2,3,4,5], Zhongtao Zhang[1,4,6✉] & Dong Zhang [1,2,3,4,5,6✉]

Hepatic inflammation is the driving force for the development and progression of NASH. Treatment targeting inflammation is believed to be beneficial. In this study, adoptive transfer of CD4$^+$ T cells converted double negative T cells (cDNT) protects mice from diet-induced liver fat accumulation, lobular inflammation and focal necrosis. cDNT selectively suppress liver-infiltrating Th17 cells and proinflammatory M1 macrophages. IL-10 secreted by M2 macrophages decreases the survival and function of cDNT to protect M2 macrophages from cDNT-mediated lysis. NKG2A, a cell inhibitory molecule, contributes to IL-10 induced apoptosis and dampened suppressive function of cDNT. In conclusion, ex vivo-generated cDNT exert potent protection in diet induced obesity, type 2 diabetes and NASH. The improvement of outcome is due to the inhibition on liver inflammatory cells. This study supports the concept and the feasibility of potentially utilizing this autologous immune cell-based therapy for the treatment of NASH.

[1] General Surgery Department, Beijing Friendship Hospital, Capital Medical University, Beijing, China. [2] Experimental and Translational Research Center, Beijing Friendship Hospital, Capital Medical University, Beijing, China. [3] Beijing Clinical Research Institute, Beijing, China. [4] Beijing Key Laboratory of Tolerance Induction and Organ Protection in Transplantation, Beijing, China. [5] Immunology Research Center for Oral and Systemic Health, Beijing Friendship Hospital, Capital Medical University, Beijing, China. [6] National Clinical Research Center for Digestive Diseases, Beijing, China. [7] Department of Gastroenterology and Hepatology, Beijing You'an Hospital, Capital Medical University, Beijing, China. ✉email: zhangzht@ccmu.edu.cn; zhangd@ccmu.edu.cn

Nonalcoholic fatty liver disease (NAFLD) is one of the most important causes of liver disease worldwide[1]. Nonalcoholic steatohepatitis (NASH) is the progressive form of NAFLD, can potentially progress to liver cirrhosis and hepatocellular carcinoma, and is associated with an increased risk of morbidity and mortality related to the liver and cardiovascular system[2,3]. Although a wealth of information on the pathogenesis of NASH has accumulated during the past 10 years, no approved therapy for NASH is available[4].

Hepatic inflammation plays a key role in the progression from hepatic steatosis to NASH, and the infiltration of different subsets of inflammatory cells is the hallmark of NASH[5]. Both innate and adaptive immune cells are involved in the pathogenesis of NASH[6-8]. Inhibition of CCR2[+] monocytes efficiently ameliorates insulin resistance, hepatic inflammation, and fibrosis in diet-induced mouse models[9]. Oral administration of anti-CD3 mAbs has the potential to suppress the inflammatory process in NASH and T2D by promoting the secretion of anti-inflammatory cytokines by Tregs[10]. These studies provide evidence that inhibiting inflammation could alleviate liver damage, and prevent the development and progression of NASH.

Double-negative T cells (DNT), which are characterized as TCRαβ[+]CD3[+]CD4[−]CD8[−]NK1.1[−] in mice and TCRαβ[+]CD3[+]CD4[−]CD8[−]CD56[−] in humans, comprise only 1–3% of peripheral T lymphocytes in mice and humans[11,12]. An increasing number of studies have proven that this rare T cell population plays critical and diverse roles in the immune system. Some reports showed that DNT were involved in systemic inflammation and tissue damage under autoimmune or inflammatory conditions[13-16]. However, studies also suggested that DNT were essential for maintaining immune system homeostasis with antigen specificity[17,18]. Young NOD mice that display a high proportion of splenic DNT are potentially resistant to diabetes[19]. Graft-versus-host disease does not develop in individuals with >1% DNT[20]. We have identified a differentiation pathway from CD4[+] T cells to DNT in vitro and in vivo[21,22]. Ex vivo-generated CD4[+] T cell-converted DNT (cDNT) could prevent or reverse the onset of type 1 diabetes, allergic asthma, and induce allograft tolerance[21,23-25].

In this study, we demonstrated that the adoptive transfer of cDNT ameliorated diet-induced NASH development. We also provide evidence that cDNT exert immunoregulatory activity primarily by suppressing proinflammatory M1 macrophages, not regulatory M2 macrophages, in NASH. The discrepant inhibition of M1/M2 macrophages by cDNT is mainly regulated by exogenous IL-10 and NKG2A expression by cDNT.

## Results

### A single transfer of cDNT prevented diet-induced obesity, insulin resistance, and NASH development.

To investigate the therapeutic effect of cDNT in NASH, we transferred cDNT to mice with diet-induced NASH (Fig. 1a). Compared with control mice, high-fat diet (HFD)-fed mice that received a single transfer of cDNT exhibited significantly less weight gain (Fig. 1b). The transferred cDNT mainly accumulated in the liver, draining lymph nodes (LN) and visceral adipose tissue (VAT) of recipient animals (Fig. 1c). Food intake was not significantly different between groups (Fig. 1d). Liver and adipose tissue weights tended to be lower in HFD-fed mice with cDNT treatment (Fig. 1e). After cell transfer, fasting glucose levels were significantly lower in HFD-fed mice at 16 weeks (Fig. 1f). cDNT transfer in HFD-fed mice also improved glucose tolerance and insulin sensitivity (Fig. 1g, h).

HFD-fed mice that received cDNT also showed decreased plasma levels of alanine transaminase (ALT), aspartate transaminase (AST), and triglycerides (TG; Fig. 1i). HFD-fed mice that received cDNT had smaller adipocytes and less lymphocyte infiltration of adipose tissue, liver fat accumulation, lobular inflammation, and focal liver necrosis (Fig. 1j, k). cDNT transfer significantly decreased the NAFLD activity score (NAS), and the staining of Sirius red and α-smooth muscle actin (α-SMA), downregulated hepatic hydroxyproline content levels in HFD-fed mice (Fig. 1j–l). Meanwhile, fibrosis-related genes, such as α-SMA, Col1a1, and Col3a1, were also downregulated in the liver tissues that underwent cDNT transfer (Fig. 1m). Mice received cDNT therapy had lower plasma IFN-γ, TNF-α, IL-6, IL-9, IL-17A, and IL-17F levels (Fig. 1n).

To further confirm the therapeutic effects of cDNT in NASH development, we also transferred cDNT into methionine/choline-deficient diet (MCD)-fed mice. As shown in Supplementary Fig. S1a, the bodyweight of MCD-fed mice was not altered by cDNT; however, adoptive transfer of cDNT markedly decreased mouse plasma ALT, AST, and TG levels (Supplementary Fig. S1b). Compared with control mice, MCD-fed mice received cDNT showed significantly decreased fat accumulation, lobular inflammation, and focal necrosis (Supplementary Fig. S1c), lowered Sirius red staining and α-SMA levels (Supplementary Fig. S1c, d). Hepatic hydroxyproline content and fibrosis-related genes, such as α-SMA, Col1a1, Col3a1, and Tgfb, also decreased in mice treated with cDNT (Supplementary Fig. S1e, f). Proinflammatory cytokines, such as TNF-α, IL-6, IL-9, IL-17F, and IL-21, were also decreased in mice treated with cDNT (Supplementary Fig. S1g).

Rapamycin, an immunosuppressant, the inhibitor of mTOR signal pathway, could also block diet-induced NAFLD[26,27]. To further evaluate the therapeutic effects of cDNT in NASH development, we used rapamycin as a positive control for the efficacy of cDNT therapy. These observations were similar with that in MCD-fed mice with rapamycin treatment. As shown in Supplementary Fig. S2a, b, cDNT and rapamycin had similar therapeutic effects on NASH. Meanwhile, rapamycin also inhibited liver-resident DNT IL-17 secretion; however, the transfer of cDNT did not influence liver-resident DNT cytokine secretion (Supplementary Fig. S2d, e). Compared with liver-resident DNT, the transferred cDNT did not secrete IL-17 (Supplementary Fig. S2f, g).

Similar results were also found in choline-deficient high-fat diet (CD-HFD)-fed mice, and the transfer of cDNT decreased mice bodyweight; fasting glucose levels; liver fat accumulation; lobular inflammation; focal liver necrosis; hepatic fibrosis; plasma ALT, AST, and TG levels; and hepatic hydroxyproline levels (Supplementary Fig. S3).

### cDNT mainly accumulated in liver and draining lymph nodes, played immunoregulatory roles in NASH mice.

In order to confirm the cDNT dynamics of distribution in vivo, we transferred CD45.1[+] cDNT into MCD-fed mice. The typical flow cytometry gating strategy to distinguish hepatic lymphocytes is shown in Supplementary Fig. S4a. As shown in Supplementary Fig. S5a, b, we detected CD45.1[+] cDNT in blood, spleen, liver, and different LN every week for 4 weeks. We found CD45.1[+] cDNT mainly accumulated in liver tissues and draining lymph nodes. CXCL9 and CXCL10 were upregulated in adipose and liver tissues of mice with diet-induced NASH (Supplementary Fig. S5c).

Then we detected the cytokine secretion and immune regulatory molecules expression of these cDNT in vivo and in vitro. As shown in Supplementary Fig. S6a–c, compared with activated CD4[+] T cells, these transferred CD45.1[+] cDNT had low or no secretion of IL-2, IL-4, IL-6, IL-10, IL-13, IL-17, IL-21, and TGF-β, except for IFN-γ. However, compared with CD4[+] T cells,

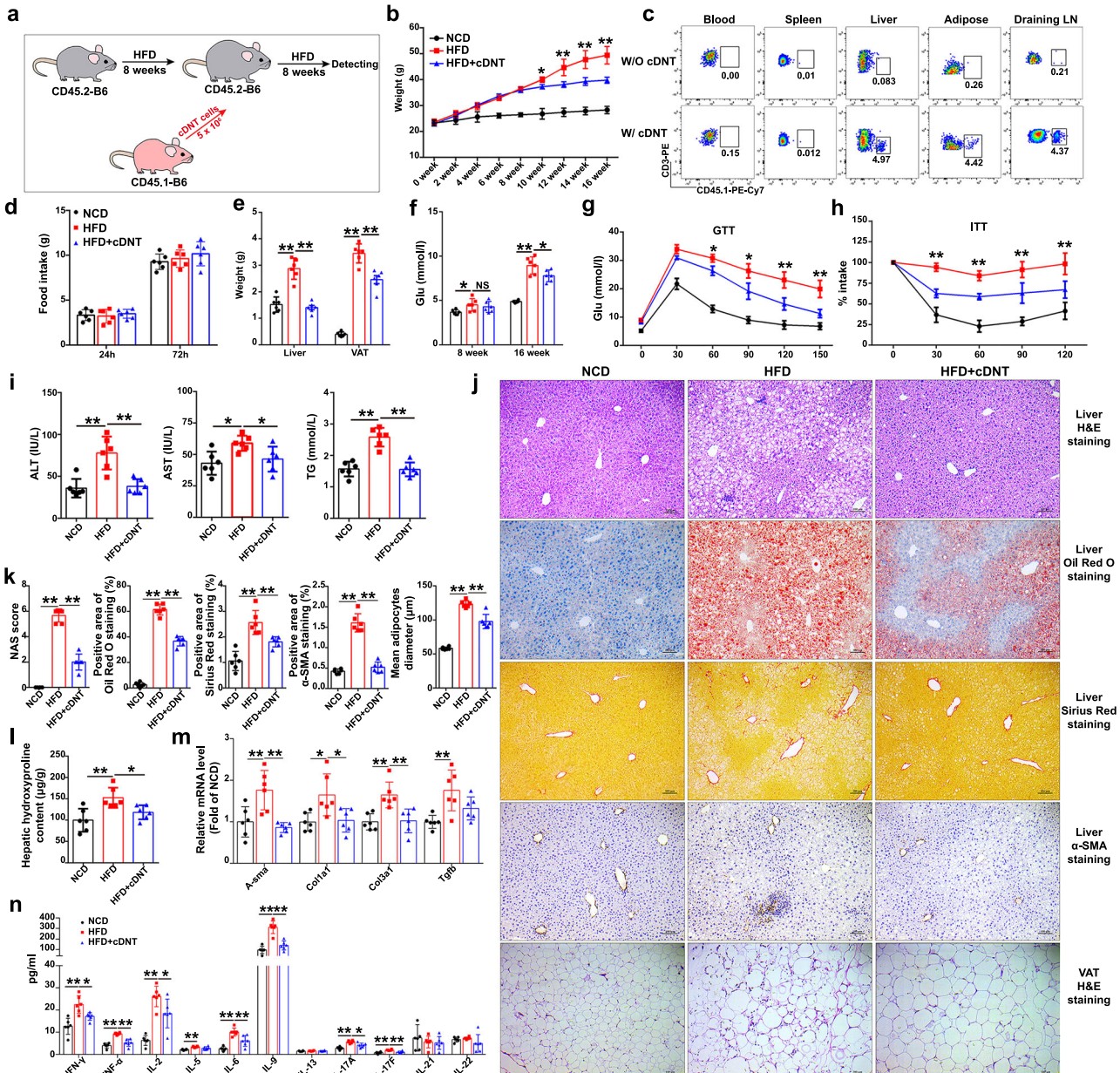

**Fig. 1 Adoptive cDNT transfer alleviated obesity-induced insulin resistance and steatohepatitis in HFD-fed mice. a** Flowchart of the adoptive transfer of cDNT. **b** Bodyweight of mice from each group. NCD, black; HFD, red; HFD + cDNT, blue. Actual *p* values (left → right): 0.012, 0.00027, 0.000038, 0.000005. **c** The in vivo distribution of transferred cDNT in mice. **d** Food intake by mice from each group. **e** Liver and VAT weights of mice fed the NCD or HFD for 16 weeks. Actual *p* values (left → right): 0.000004, 0.000001, 0.000004, 0.000001. **f** Fasting plasma glucose levels were measured in mice fed the HFD for 8 or 16 weeks. Actual *p* values (left → right): 0.048, 0.78, 9.69e−8, 0.025. **g**, **h** GTTs and ITTs were performed after intraperitoneal glucose or insulin injection. Actual *p* values (left → right): **g** 0.037, 0.015, 0.0041, 0.0031; **h** 8.58e−8, 2.74e−7, 0.000029, 0.000097. **i** Plasma ALT, AST, and TG levels were measured. Actual *p* values (left → right): 0.00030, 0.00049, 0.015, 0.045, 0.000013, 0.000011. **j** Representative H&E staining, oil red o staining, Sirius red staining, α-SMA staining in liver paraffin sections, and H&E staining in VAT paraffin sections. Scale bars, 100 μm (*n* = 6 biologically independent samples per group). **k** Quantification of liver and VAT histology staining. Actual *p* values (left → right): 0.00065, 0.00028, 5.80e−9, 1.22e−8, 0.000010, 0.0074, 8.30e−9, 1.42e−8, 5.93e−9, 0.000029. **l** Hydroxyproline levels in liver tissues of each group. Actual *p* values (left → right): 0.0032, 0.048. **m** Fibrosis-related genes in liver tissues. Actual *p* values (left → right): 0.0052, 0.0014, 0.017, 0.024, 0.0021, 0.0030, 0.0040. **n** Plasma cytokines of mice in each group. Actual *p* values (left → right): 0.00027, 0.029, 0.000002, 0.000055, 0.000009, 0.035, 0.00069, 0.000009, 0.0049, 0.000003, 0.000036, 0.000039, 0.010, 0.00014, 0.0018. Data are presented as the mean ± SD; *n* = 6 mice/group. Statistical analysis for the first left figure in **k** (NAS score group) was performed by Kruskal–Wallis multiple comparisons test, and others were performed by one-way ANOVA with post hoc multiple comparisons test. Two-sided *p* values < 0.05 were considered significant. \**p* < 0.05; \*\**p* < 0.01. Source data, including exact *p* values, are provided as a Source data file.

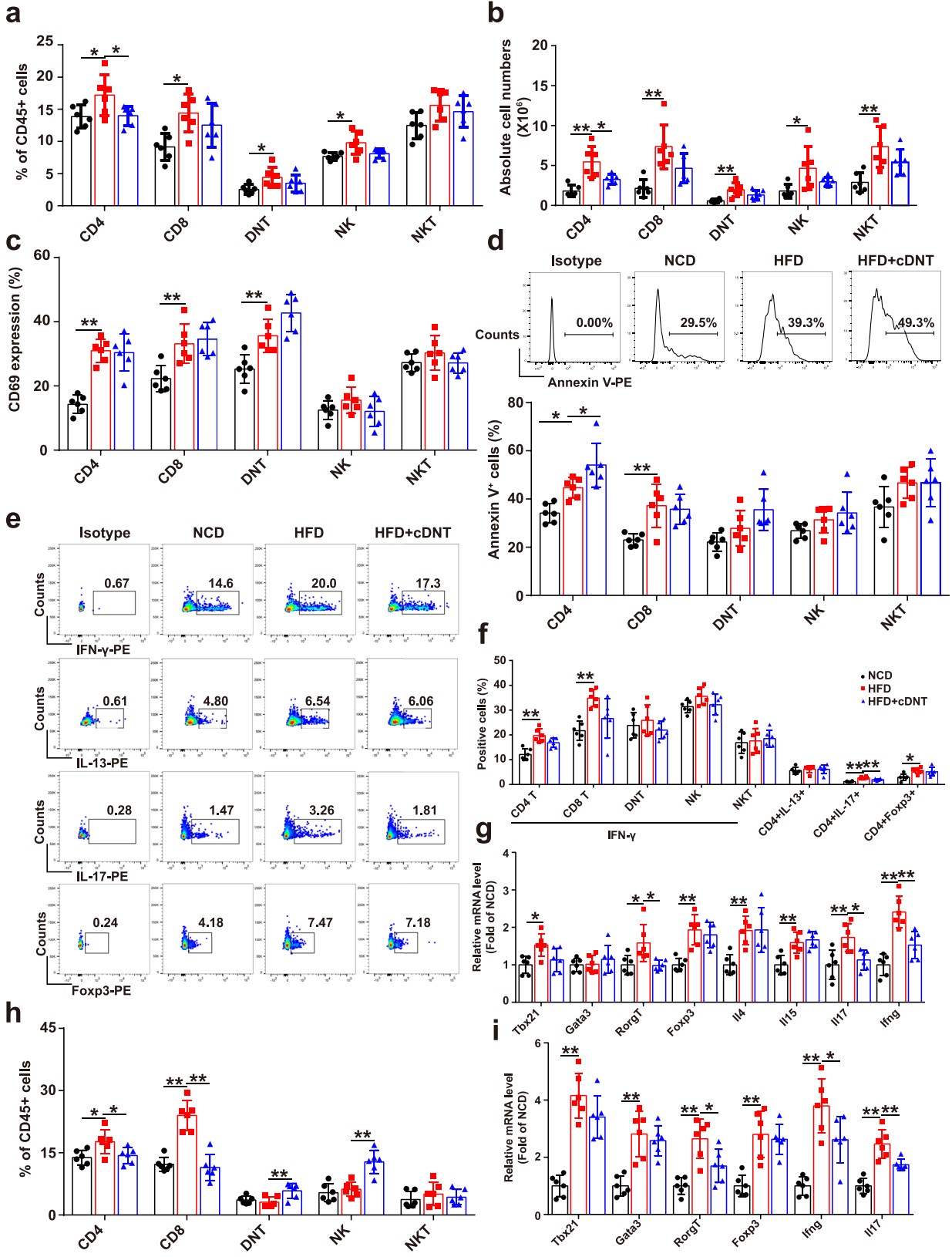

the converted cDNT highly expressed Granzyme B (GZMB), the key molecular of regulatory cDNT. Furthermore, we also detected the ability of cDNT to produce cytokines in vivo. As shown in Supplementary Fig. S6d, these cDNT also had low or no secretion of IL-2, IL-4, IL-6, IL-10, and IL-17, including IFN-γ in vivo, but

highly expressed GZMB. These observations indicated that these transferred cDNT were regulatory immune cells and were not proinflammatory cDNT.

Meanwhile, the apoptosis and proliferation of CD45.1$^+$ cDNT in liver tissues were also detected. As shown in Supplementary

**Fig. 2 cDNT transfer decreased the CD4$^+$ T cell proportion and survival, and Th17 differentiation in liver and adipose tissue. a** Percentages of intrahepatic CD4$^+$ T cells, CD8$^+$ T cells, DNT, NK cells, and NKT cells relative to the total number of CD45.2$^+$ cells in mice. NCD, black; HFD, red; HFD + cDNT, blue. Actual $p$ values (left → right): 0.028, 0.027, 0.017, 0.033, 0.016. **b** Absolute numbers of CD4$^+$ T cells, CD8$^+$ T cells, DNT, NK cells, and NKT cells in mouse livers. Actual $p$ values (left → right): 0.00039, 0.021, 0.0039, 0.0045, 0.026, 0.0027. **c** Statistical analysis of the percentages of CD69$^+$ cells among T cells, NK cells, and NKT cells. Actual $p$ values (left → right): 0.000019, 0.0060, 0.0094. **d** Representative flow cytometry plots (top) and statistical analysis (bottom) of Annexin V$^+$ cells relative to total T cells, NK cells, and NKT cells in each group. Actual $p$ values (left → right): 0.027, 0.047, 0.0045. **e, f** Representative flow cytometry plots (**e**) and statistical analysis (**f**) of the percentages of IFN-γ$^+$, IL-13$^+$, IL-17$^+$, and Foxp3$^+$ cells among liver infiltrated T cells. Actual $p$ values (left → right): **f** 0.00013, 0.00013, 0.000006, 0.0081, 0.015. **g** Real-time PCR analysis of related genes in livers from mice in each group. Actual $p$ values (left → right): 0.015, 0.016, 0.012, 0.00041, 0.0066, 0.0029, 0.0063, 0.024, 0.000016, 0.0017. **h** Percentages of T cells, NK cells, and NKT cells among CD45.2$^+$ cells from adipose tissue. Actual $p$ values (left → right): 0.021, 0.034, 0.000013, 0.000006, 0.0092, 0.00042. **i** Relative mRNA levels of *Tbx21*, *Gata3*, *Foxp3*, *RORγt*, *Ifng*, and *Il17* in adipose tissue. Actual $p$ values (left → right): 0.000038, 0.00022, 0.00027, 0.022, 0.00028, 0.000026, 0.037, 0.000007, 0.0065. Data are presented as the mean ± SD; $n = 6$ mice/group. Statistical analysis for **b** of CD8 group, **d** of DNT group, and **g** of *Rorgt* group was performed by Kruskal–Wallis multiple comparisons test, and others were performed by one-way ANOVA with post hoc multiple comparisons test. Two-sided $p$ values < 0.05 were considered significant. \*$p$ < 0.05; \*\*$p$ < 0.01. Source data, including exact $p$ values, are provided as a Source data file.

Fig. S6e, f, the percentage of Annexin V$^+$ cDNT were <15%, and proportion of Ki67$^+$ cDNT were >40% at 3 and 4 weeks after cDNT adoptive transfer, which indicated that the transferred cDNT had low apoptosis and could proliferate in liver tissues. Meanwhile, these cDNT has also maintained a CD4$^-$CD8$^-$, suggesting these cells were stable in vivo (Supplementary Fig. S6g).

**cDNT transfer decreased the CD4$^+$ T cell proportion, survival and differentiation in liver, and adipose tissue of NASH mice.** To elucidate the mechanisms by which cDNT prevent NASH development, we studied the lymphocytes composition in each group. As shown in Fig. 2a, b and Supplementary Fig. S7a–c, the transferred cDNT markedly reduced the proportion of CD4$^+$ T cells, but not of CD8$^+$, NK, and NKT cells in liver tissues, spleen, and draining LN. Immunohistochemistry staining also showed the transferring of cDNT could decrease liver-infiltrating CD45$^+$, CD3$^+$, but not CD20$^+$ cells (Supplementary Fig. S7d, e). Although these cDNT did not influence hepatic T cell activation in HFD-fed mice (Fig. 2c), the proportion of apoptotic CD4$^+$ T cells was increased (Fig. 2d). Among CD4$^+$ T cells, those producing IFN-γ and IL-17 were more abundant in HFD-fed mice, while the corresponding mice that received cDNT exhibited a marked decrease in the percentage of IL-17-producing CD4$^+$ T cells only. These cDNT did not lower the proportion of Foxp3$^+$ cells among total CD4$^+$ T cells (Fig. 2e, f). Furthermore, HFD-fed mice showed considerably upregulation of some Th lineage-defining transcription factors (*Tbx21*, *RORγt*, and *Foxp3* but not *Gata3*), while cDNT transfer significantly decreased the levels of *Tbx21* and *RORγt*. Meanwhile, the mRNA levels of some proinflammatory cytokines in liver tissue, such as *Ifng* and *Il17*, were also downregulated in cDNT-treated mice (Fig. 2g). Similar results were also found in adipose tissue, cDNT transfer significantly decreased CD4$^+$ and CD8$^+$ T cell proportions (Fig. 2h). The mRNA levels of the transcription factor *RORγt*, and the proinflammatory cytokines *Il17* and *Ifng* were also downregulated in adipose tissue after cDNT transfer (Fig. 2i). In order to further confirm the suppression of these transferred cDNT to Th17 cells, we cocultured the cDNT and Th17 cells in vitro. As shown in Supplementary Fig. S7f, g, cDNT could not only increase the apoptosis of CD4$^+$ T cells, including Th17 cells, but also suppress the IL-17 secretion of CD4$^+$ T cells. These observations indicated that a single transfer of cDNT to HFD-fed mice could reduce hepatic inflammation, and NASH development by decreasing hepatic CD4$^+$ T cell proportion and survival, and Th17 cell differentiation.

**Transferred cDNT suppressed liver-infiltrating monocyte-derived macrophages, especially M1 macrophages.** In this study, we found HFD-fed mice had an increased proportion of

liver-infiltrating macrophages, and cDNT significantly inhibited macrophage liver infiltration (Fig. 3a). However, no significant differences in the proportions of Kupffer cells, neutrophils, or dendritic cells (DCs) were found in HFD-fed mice with or without cDNT transfer. The flow cytometry gating strategy to distinguish hepatic innate immune cells is shown in Supplementary Fig. S4b. Immunohistochemistry staining of CD11b and F4/80 in liver tissues of HFD-fed mice also showed cDNT transfer lowered liver-infiltrating CD11b and F4/80-positive cells (Supplementary Fig. S8a). Furthermore, we quantified the proportions of proinflammatory M1 (CD11C$^+$CD206$^-$) and anti-inflammatory M2 (CD11C$^-$CD206$^+$) macrophages in the liver. As shown in Fig. 3b–d, HFD-fed mice had a higher frequency and number of M1 macrophages, and cDNT significantly decreased the proportions of these cells. Meanwhile, we also defined hepatic M1 macrophages and M2 macrophages by MHC II$^+$CD206$^-$ and MHC II$^-$CD206$^+$, respectively, similar results were also found in Fig. 3b, e. Ly6C$^{hi}$ monocytes are the main proinflammatory macrophages that infiltrate the liver during NASH development, and transferred cDNT could also markedly decrease the proportion of these cells in HFD-fed mice. Interestingly, cDNT did not influence the M2 macrophage proportion. Meanwhile, cDNT remarkably decreased macrophage TNF-α and IL-6 secretion in HFD (Fig. 3f, g). However, Kupffer cells was not significantly affected by cDNT. As shown in Fig. 3h, cDNT markedly decreased the mRNA expression of the M1 macrophage markers *iNOS*, *H2-DMb1/2*, *H2-IA*, *Tlr4*, *Ccr2*, and *Tnfa*, but did not affect the expression of the M2 macrophage marker *Arg-1* in HFD-fed mice. Similar results were obtained in adipose tissue; transferred cDNT suppressed macrophage infiltration and M1 polarization in HFD-fed mice (Fig. 3i, j and Supplementary Fig. S8b). The mRNA levels of *F4/80*, *iNOS*, and *Tnfa* were also significantly downregulated after cDNT transfer, but *Arg-1* and *Il10* mRNA levels in adipose tissue were not significantly affected by cDNT in HFD-fed mice (Fig. 3k).

The same analyses were performed in MCD- or CD-HFD-fed mice with or without cDNT transfer, and similar results were obtained (Supplementary Figs. S9 and S10). These results suggested that the improvement of liver steatosis by cDNT transfer is associated with decreased recruitment of infiltrating monocyte-derived macrophages, especially M1 macrophages, and proinflammatory cytokine secretion.

**cDNT-mediated cytotoxicity against M2 macrophages compared with M1 macrophages was impaired due to IL-10.** In this study, transferred cDNT could enhance the apoptosis of hepatic M1 macrophages, but not M2 macrophages in mice (Fig. 4a). In vitro coculture study also revealed that cDNT markedly induced M1 macrophage apoptosis, which was associated with decreased

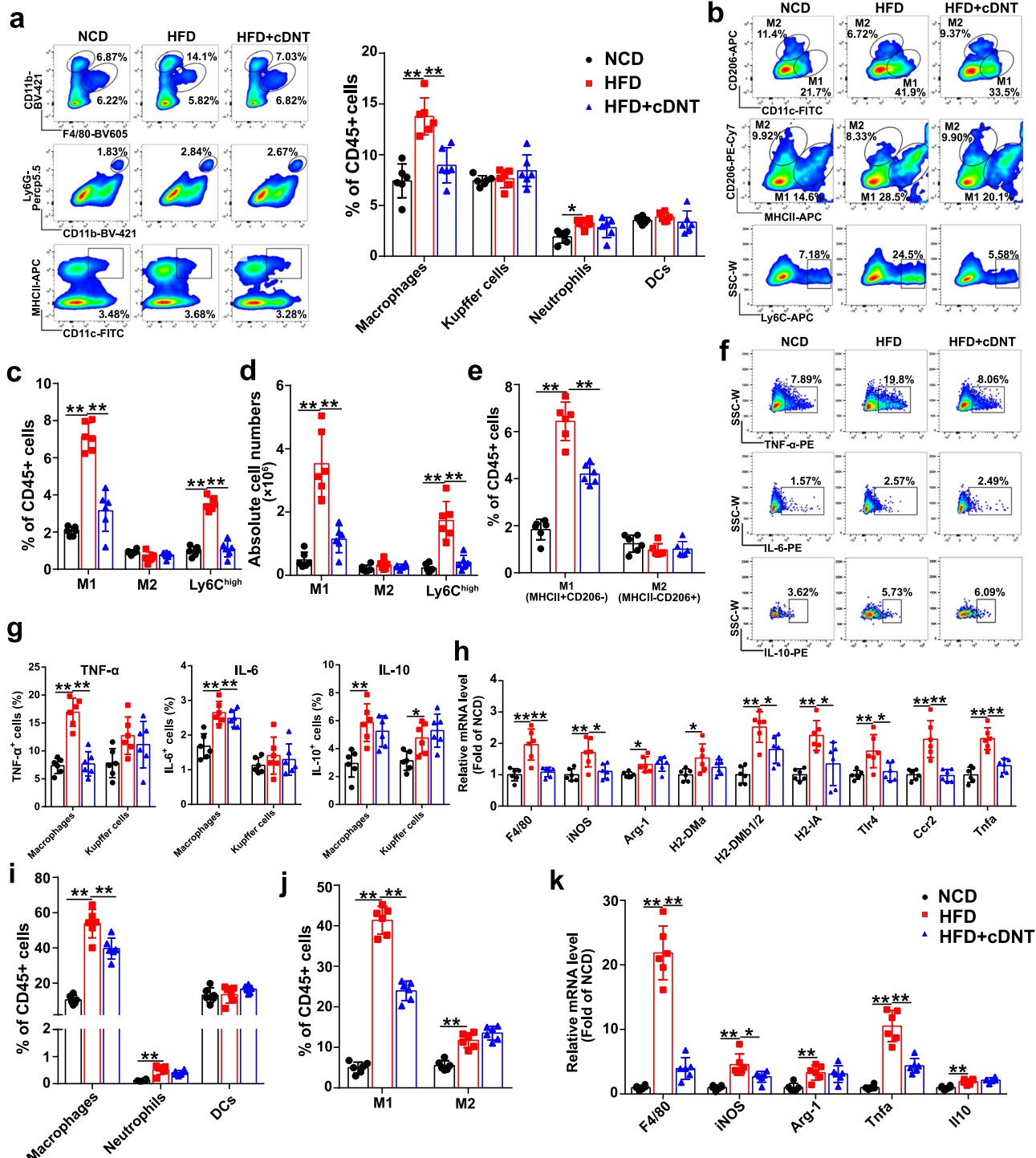

TNF-α and IFN-γ secretion by macrophages; however, no enhanced apoptosis was observed in M2 macrophages (Fig. 4b, c).

Then, cDNT were pre-stimulated with M1 macrophage cytokines (IFN-γ, IL-6, and TNF-α), M2 macrophage cytokines (IL-4 and IL-10), and IL-17 for 24 h and then cocultured with macrophages for another 24 h. As shown in Fig. 4d, only IL-10 decreased cDNT-mediated macrophage lysis. Anti-IL-10 mAbs significantly increased the cDNT-mediated lysis of M2 macrophages (Fig. 4e). These observations indicated that IL-10 secreted by M2 macrophages might protect M2 cells from cDNT-mediated lysis by downregulating the suppressive functions of cDNT.

To better understand the influence of IL-10 on cDNT function, we also stimulated cDNT with different concentrations of IL-10. As shown in Fig. 4f, IL-10 induced cDNT apoptosis and inhibited cDNT proliferation in a dose-dependent manner. We also examined cytokine receptors in cDNT via real-time PCR and found that the highly expressed genes in cDNT were *Il4r*, *Il10r*, and *Ifngr* (Fig. 4g). As shown in Fig. 4h, there was a dose-dependent increase in IL-10 receptor (IL-10R) expression in cDNT after IL-10 stimulation. Moreover, the IL-10-induced inhibition of cDNT survival and proliferation was abrogated by the anti-IL-10R neutralizing mAb (Fig. 4i).

**Fig. 3 Transferred cDNT inhibited the proportion of infiltrating macrophages, especially M1 macrophages, and the secretion of TNF-α in liver and adipose tissue. a** Representative flow cytometry plots (left) and statistical analysis (right) of macrophages, Kupffer cells, neutrophils, and DCs relative to the total intrahepatic CD45.2$^+$ cells in mice. NCD, black; HFD, red; HFD + cDNT, blue. Actual p values (left → right): 0.000038, 0.00065, 0.013. **b** Flow cytometric images of M1 and M2 macrophages and Ly6C$^{high}$ cells. **c** Statistical analysis of the percentages of M1, M2, and Ly6C$^{high}$ macrophages among liver-infiltrating macrophages. Actual p values (left → right): 3.91e−8, 8.05e−7, 1.42e−8, 2.23e−8. **d** Absolute numbers of M1, M2, and Ly6C$^{high}$ macrophages in the liver. Actual p values (left → right): 0.000003, 0.000046, 0.000015, 0.000062. **e** Statistical analysis of the percentages of M1 (MHC II$^+$CD206$^−$) and M2 (MHC II$^−$CD206$^+$) macrophages among liver-infiltrating CD45$^+$ cells. Actual p values (left → right): 8.05e−9, 0.000024. **f, g** Representative flow cytometry plots (**f**) and statistical analysis (**g**) of the percentages of TNF-α$^+$, IL-6$^+$, and IL-10$^+$ cells among macrophages and Kupffer cells. Actual p values (left → right): **g** 0.000003, 0.000004, 8.02e−9, 0.000024, 0.0015, 0.034. **h** Relative mRNA levels of the indicated genes in livers. Actual p values (left → right): 0.00029, 0.00068, 0.0041, 0.014, 0.030, 0.025, 0.000036, 0.023, 0.0015, 0.016, 0.0060, 0.016, 0.00024, 0.00020, 0.000008, 0.00020. **i, j** Percentages of macrophages, neutrophils, DCs, and M1 and M2 macrophages among CD45.2$^+$ cells in adipose tissue. Actual p values (left → right): **i** 1.27e−8, 0.0026, 0.000059; **j** 5.80e−9, 2.05e−8, 0.000019. **k** Relative mRNA levels of the indicated genes in adipose tissue. Actual p values (left → right): 7.29e−9, 1.82e−8, 0.00013, 0.022, 0.0052, 7.18e−8, 0.000016, 0.0017. Data are presented as the mean ± SD; n = 6 mice/group. Statistical analysis was performed by one-way ANOVA with post hoc multiple comparisons test. Two-sided p values < 0.05 were considered significant. *p < 0.05; **p < 0.01. Source data, including exact p values, are provided as a Source data file.

These observations indicated that IL-10 secreted by M2 macrophages could decrease cDNT survival and function to protect M2 macrophages from cDNT-mediated lysis.

**Transcriptome sequencing analysis showed that IL-10 inhibited cDNT survival and cytotoxicity, and enhanced the expression of the inhibitory molecule NKG2A.** To further understand the role of IL-10 in the regulation of cDNT survival and function, cDNT treated with or without IL-10 for 24 h were compared in a transcriptome sequencing study (Fig. 5a). Gene ontology (GO) and Kyoto Encyclopedia of Genes and Genomes (KEGG) pathway analyses revealed that the regulated genes in IL-10-stimulated cDNT were involved in the immune response, inflammatory response, regulation of apoptosis and proliferation, cytolysis, cytokine–cytokine receptor interaction, and T cell receptor signaling pathway, among others (Fig. 5b, c).

Compared with control cDNT, IL-10-treated cDNT showed the downregulation of cell survival and activation genes (*Bcl-xl*, *Epcam*, and *Spi1*), cell cytotoxicity genes (*Nkp46*, *Klrg1*, *Slamf6*, *Cd107a*, *Cd161*, and *Cd226*), and immune response genes (*Lta*, *Cd127*, *H2-Oa*, and *H2-Dma*; Fig. 5d). We also demonstrated these changes in related genes through real-time PCR and flow cytometry analysis. As shown in Fig. 5e, f, IL-10 downregulated Bcl-2 and Bcl-xL expression in cDNT, while significantly enhanced Cytc and Cflar expression, which indicated that IL-10 regulated cDNT survival through these molecules. Although perforin and FasL are the major functional molecules of cDNT, we did not observe a significant difference in *Prf1* or *Fasl* expression between IL-10-treated and untreated cDNT. And IL-10 stimulation did not affect the cDNT cytokine secretion (Supplementary Fig. S11c). However, IL-10 significantly increased *Nkg2a* mRNA level in cDNT (Fig. 5g). Moreover, IL-10 markedly upregulated NKG2A protein expression in cDNT, but did not affect NKG2D expression (Fig. 5h). Moreover, anti-IL-10R mAbs significantly reduced NKG2A expression in IL-10-stimulated cDNT (Fig. 5i). These observations demonstrated that IL-10 suppressed cDNT cytotoxicity by promoting NKG2A expression.

The phosphatidylinositol 3-kinase (PI3K) and mitogen-activated protein kinase (MAPK) signaling pathways play important roles in NKG2A expression[28]. In this study, we found that IL-10-stimulated cDNT highly expressed PI3K-AKT signaling genes, such as *Jak3*, *Pik3ap1*, *Pik3cb*, and *Sgk1*, and MAPK pathway genes, such as *Stmn1*, *Fos*, *Kras*, *Nras*, and *Map3k5*, which may be involved in the IL-10-mediated upregulation of NKG2A expression in cDNT (Fig. 5d).

mTOR signal pathway plays important roles in regulating the DNT activation and function. However, as shown in Supplementary Fig. S11a–c, IL-10 stimulation did not increase pi-mTOR expression, and rapamycin stimulation did not influence cDNT cytokine secretion, except IFN-γ, which indicated mTOR signal pathway might not important for cDNT function.

**IL-10 secreted by M2 macrophages inhibited cDNT-mediated cytolysis by increasing the NKG2A/Qa-1b interaction.** To further confirm the upregulation of NKG2A expression by cDNT in response to IL-10 secreted by M2 macrophages, we first cocultured cDNT with M1 or M2 macrophages. As shown in Fig. 6a, b, NKG2A expression on cDNT markedly increased following exposure to M2 macrophages, not M1 macrophages. As expected, neutralizing M2 macrophage secreted IL-10 by anti-IL-10 antibody led to the downregulation of NKG2A expression in cDNT (Fig. 6c).

We also found that the NKG2A ligand Qa-1b was highly expressed in M1 and M2 macrophages (Fig. 6d). When we incubated M1 or M2 macrophages with saturating amounts of anti-Qa-1b mAb to block the interaction of Qa-1 with its receptor CD94/NKG2A, the cytotoxicity of cDNT against M2 macrophages was significantly enhanced, while the cytotoxicity of cDNT against M1 macrophages did not change markedly (Fig. 6e).

Furthermore, as shown in Fig. 6f, although IL-10 significantly attenuated the cDNT-mediated lysis of macrophages, the IL-10-mediated inhibition of cDNT cytolysis was remarkably abrogated by anti-Qa-1b neutralizing mAb. These observations demonstrated that IL-10 secreted by M2 macrophages could inhibit cDNT-mediated cytolysis by upregulating the NKG2A/Qa-1b interaction.

In order to further confirm the roles of IL-10 in regulating cDNT cytotoxicity, we treated the MCD-fed mice with anti-IL-10 neutralizing antibody after cDNT transfer for 4 weeks. As shown in Fig. 7a–c, mice treated with IL-10 neutralizing antibody together with cDNT showed increased plasma ALT levels, liver fat accumulation, inflammation, and NASH development. Meanwhile, we also detected the proportion of M1 and M2 macrophages after anti-IL-10 neutralizing antibody treatment. Compared with cDNT treatment alone, combination cDNT with anti-IL-10 neutralizing antibody decreased liver macrophages proportions, especially M2 macrophages after 4 weeks MCD feeding (Fig. 7d–f). Transferred cDNT with anti-IL-10 neutralizing antibody could also increase M2 macrophages apoptosis and inhibit cell proliferation at 4 weeks (Fig. 7g, h). IL-10 neutralizing antibody injection could inhibit Bcl-2 and Bcl-xl expression in M2 macrophages at 4 weeks (Fig. 7i, j). Furthermore, we also observed that IL-10 neutralizing antibody could downregulate NKG2A, but not NKG2D expression in cDNT at 4 weeks (Fig. 7k). These data demonstrated that anti-IL-10 neutralizing antibody increased cDNT cytotoxicity toward M2 macrophages through inhibiting cDNT NKG2A expression in vivo.

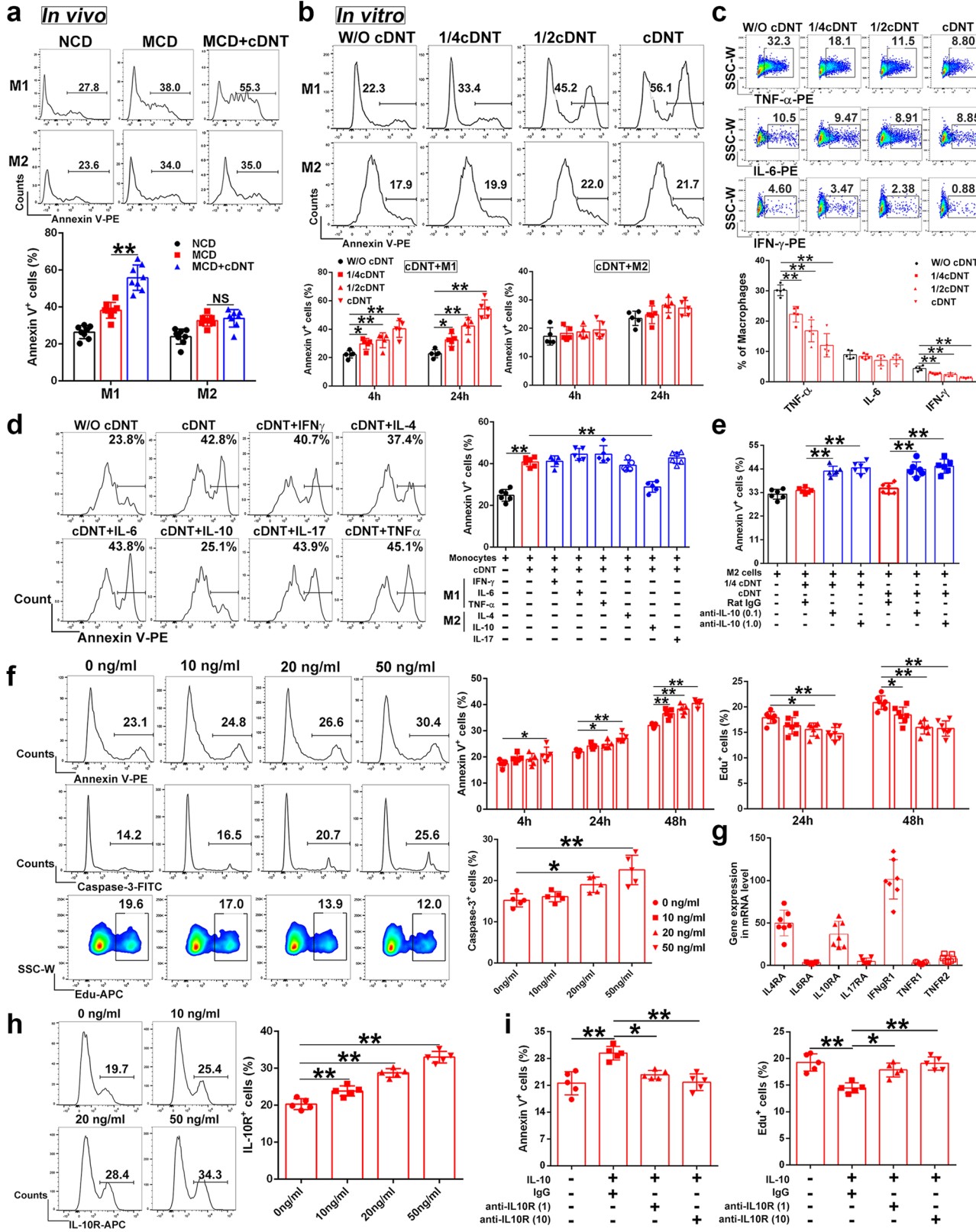

## Discussion

Accumulating evidence suggests that hepatic inflammation plays a key role in the progression from hepatic steatosis to NASH. Innovative strategies that target liver inflammation may have the benefit of preventing NASH development and improving the prognosis of NAFLD patients.

In this study, the adoptively transferred ex vivo-generated cDNT significantly inhibited liver and adipose tissue inflammation. The cDNT-treated mice exhibited significantly less weight gain, lower fasting glucose levels, and improved glucose tolerance and insulin sensitivity. cDNT also decreased plasma ALT levels, liver fat accumulation, lobular inflammation, and liver fibrosis-

**Fig. 4 M2 macrophages secreted IL-10 to decrease the suppressive functions of cDNT against M2 cells. a** Representative flow cytometry images showing the percentages of Annexin V[+] cells among M1 and M2 macrophages in liver (top). Statistical analysis of Annexin V[+] cells relative to hepatic M1 and M2 macrophages (bottom, n = 8 mice/group). NCD, black; MCD, red; MCD + cDNT, blue. Actual p values (left → right): 0.000002, 0.79. **b** CD45.1-positive M1 or M2 macrophages were cocultured with cDNT at different ratio for 4 or 24 h in vitro. Macrophage apoptosis were detected by flow cytometry (n = 5 biologically independent samples per group). W/O cDNT, black; W/cDNT, red. Actual p values (left → right): 0.036, 0.0095, 0.000029, 0.041, 0.00011, 2.57e−7. **c** CD45.1-positive bone marrow-derived macrophages were cocultured with cDNT at different ratio for 24 h in vitro. Macrophages TNF-α, IL-6, and IFN-γ secretion were detected by flow cytometry (n = 5 biologically independent samples per group). W/O cDNT, black; W/ cDNT, red. Actual p values (left → right): 0.0045, 0.00002, 3.92e−7, 0.00048, 0.000053, 1.85e−7. **d** Macrophages were pretreated with different cytokines (20 ng/ml) for 4 h, and cocultured with cDNT for 24 h. Statistical analysis of Annexin V[+] cells relative to macrophages in each group (n = 6 biologically independent samples per group). W/O cDNT, black; W/ cDNT, red; W/cytokines pretreated cDNT, blue. Actual p values (left → right): 3.54e−10, 9.30e−7. **e** M2 macrophages were pretreated with anti-IL-10 neutralizing mAb (0.1 and 1.0 μg/ml) or IgG control for 4 h, and cocultured with cDNT for 24 h. Statistical analysis of Annexin V[+] cells relative to M2 macrophages in each group (n = 6 biologically independent samples per group). W/O cDNT, black; W/ cDNT, red; W/anti-IL-10 mAb treated cDNT, blue. Actual p values (left → right): 0.000016, 7.57e−7, 0.000019, 0.000001. **f** cDNT incubated with anti-CD3/CD28 antibodies were stimulated with IL-10. cDNT apoptosis and proliferation were detected (n ≥ 5 biologically independent samples per group). Actual p values (left → right): for Annexin V[+] cells, 0.044, 0.021, 0.000081, 0.0012, 0.000041, 8.80e−7; for Edu[+] cells, 0.017, 0.00097, 0.021, 8.00e−6, 0.000004; for Caspase-3[+] cells, 0.044, 0.00038. **g** The expression of cytokine receptors in cDNT were detected through real-time PCR (n = 7 biologically independent samples per group). **h** Representative flow cytometry (left) and statistical analysis (right) of IL-10R[+] cells relative to cDNT after stimulation with IL-10 and anti-CD3/CD28 antibodies for 24 h (n = 5 biologically independent samples per group). Actual p values (left → right): 0.00057, 0.000002, 3.23e−9. **i** cDNT incubated with anti-CD3/CD28 antibodies were pretreated with anti-IL-10R neutralizing mAb (1 and 10 μg/ml) or IgG control for 4 h, and cocultured with IL-10 for 24 h. Statistical analysis of Annexin V[+] and Edu[+] cells relative to cDNT in each group (n = 5 biologically independent samples per group). Actual p values (left → right): for Annexin V[+] cells, 0.0013, 0.035, 0.0018; for Edu[+] cells, 0.0012, 0.037, 0.0019. Data are presented as the mean ± SD. Statistical analysis was performed by one-way ANOVA with post hoc multiple comparisons test. Two-sided p values < 0.05 were considered significant. *p < 0.05; **p < 0.01; NS not significant. Source data, including exact p values, are provided as a Source data file.

related gene expression. The protection from liver inflammation mediated by cDNT occurred through the selective inhibition of M1 macrophages and Th17 cells.

The DNT paly critical and diverse roles in the immune system, and they are reported to have both proinflammatory and anti-inflammatory functions. In patients with systemic lupus erythematosus (SLE)[13–15], primary Sjögren's syndrome (pSS)[16,29], and psoriasis[30], DNT that expands in peripheral blood and inflamed tissues are the major producers of the proinflammatory cytokine IL-17 and are believed to be pathogenic in these autoimmune diseases. However, the cDNT we used in this study, had low or no secretion of IL-17, highly expressed GZMB, which were defined as one type of regulatory T cells, although no Foxp3 expression. These cDNT could prevent or reverse the onset of type 1 diabetes, allergic asthma, liver injury, and induce allograft tolerance[21–25].

In our study, we found that the transferred cDNT mainly accumulated in the liver and adipose tissue of recipient animals. In our previous study, we found that cDNT expressed high levels of CXCR3[22]. In current study, we found CXCL9 and CXCL10, the ligands of CXCR3, were upregulated in liver tissue and VAT of HFD-fed mice, which may explain the cDNT liver and adipose tissue migration.

M1 macrophages and Th17 cells are major contributors to liver inflammation in NASH development[31,32]. Our study showed that cDNT transfer remarkably decreased hepatic Th17 cell proportion and survival. We also demonstrated that the transferred cDNT could suppress the liver infiltration of M1 macrophages and proinflammatory cytokine secretion. The suppression of M1 macrophages and Th17 cells by cDNT might be the major mechanism by which cDNT transfer prevent NASH development.

In NASH models, we noticed that cDNT mainly suppressed M1, not M2, macrophages in the liver. This discrepant inhibition of M1/M2 macrophages by cDNT was the most interesting phenomenon in this study. By stimulating cDNT with different cytokines secreted by M1 macrophages or M2 macrophages, we revealed that IL-10, not the other cytokines, affected cDNT inhibitory function. An in vitro study from Zhang's group reported that the administration of exogenous IL-10 rendered DN

Treg cells susceptible to TCR cross-linking-induced apoptosis[33]. In our study, we provide evidence that in addition to increased apoptosis and decreased proliferation, upregulated expression of the inhibitory molecule NKG2A might be a major contributor to IL-10-mediated decrease in cDNT function.

NKG2A is a cell surface molecule that is typically expressed by NK cells, its expression can also be induced on CD8[+] T cells. The NKG2A/CD94 complex binds to the nonclassical MHC I molecule HLA-E in humans, and Qa-1b in mice and transduces inhibitory signals, which suppress the effector function and survival of NK and CD8[+] T cells[34,35]. In addition, cytokines such as IL-15, TGF-β, IL-10, and IL-12 can play a role in CD94/NKG2 expression[35]. Qa-1 is expressed by most cell types. The Qa-1-CD94/NKG2A interaction is critical for preventing NK cell-mediated killing of DCs, B cells, and T cells[36–38]. In this study, when we added an anti-IL-10 neutralizing antibody to the M2 macrophage/cDNT coculture system, NKG2A expression in cDNT was significantly downregulated, and the cDNT-mediated lysis of M2 macrophages increased. Nevertheless, blocking the interaction of Qa-1 with its receptor CD94/NKG2A significantly enhanced the cytotoxicity of cDNT against M2 macrophages. Furthermore, the inhibition of cDNT cytolysis induced by IL-10 was remarkably abrogated by anti-Qa-1b neutralizing mAb. Therefore, the secretion of IL-10 by M2 macrophages, which induces cDNT NKG2A expression, might be the major contributor to protect against cDNT-mediated killing.

Activation of mTOR signal pathway plays important roles in liver diseases, and mTOR inhibitor rapamycin could ameliorate liver fibrosis, autoimmune hepatitis through regulation Th17/Treg cell balance[39–41]. The hepatic Foxp3[+] Tregs in rapamycin-treated mice had significantly higher frequency and suppressive effects. Activated Treg with rapamycin treatment preferentially phosphorylated STAT5 and STAT3, and did not utilize the PI3K/mTOR pathway[42,43]. In SLE patients, mTORC1 activity is prominently increased in DNT, and plays important role in DNT pro-inflammation[13,44]. In this study, IL-10 stimulation did not increase pi-mTOR expression of cDNT; however, IL-10 upregulated PI3K-AKT and MAPK pathway-related gene expression, which indicated that these signaling pathways may be involved in the IL-10-mediated cDNT function. The mechanisms of IL-10

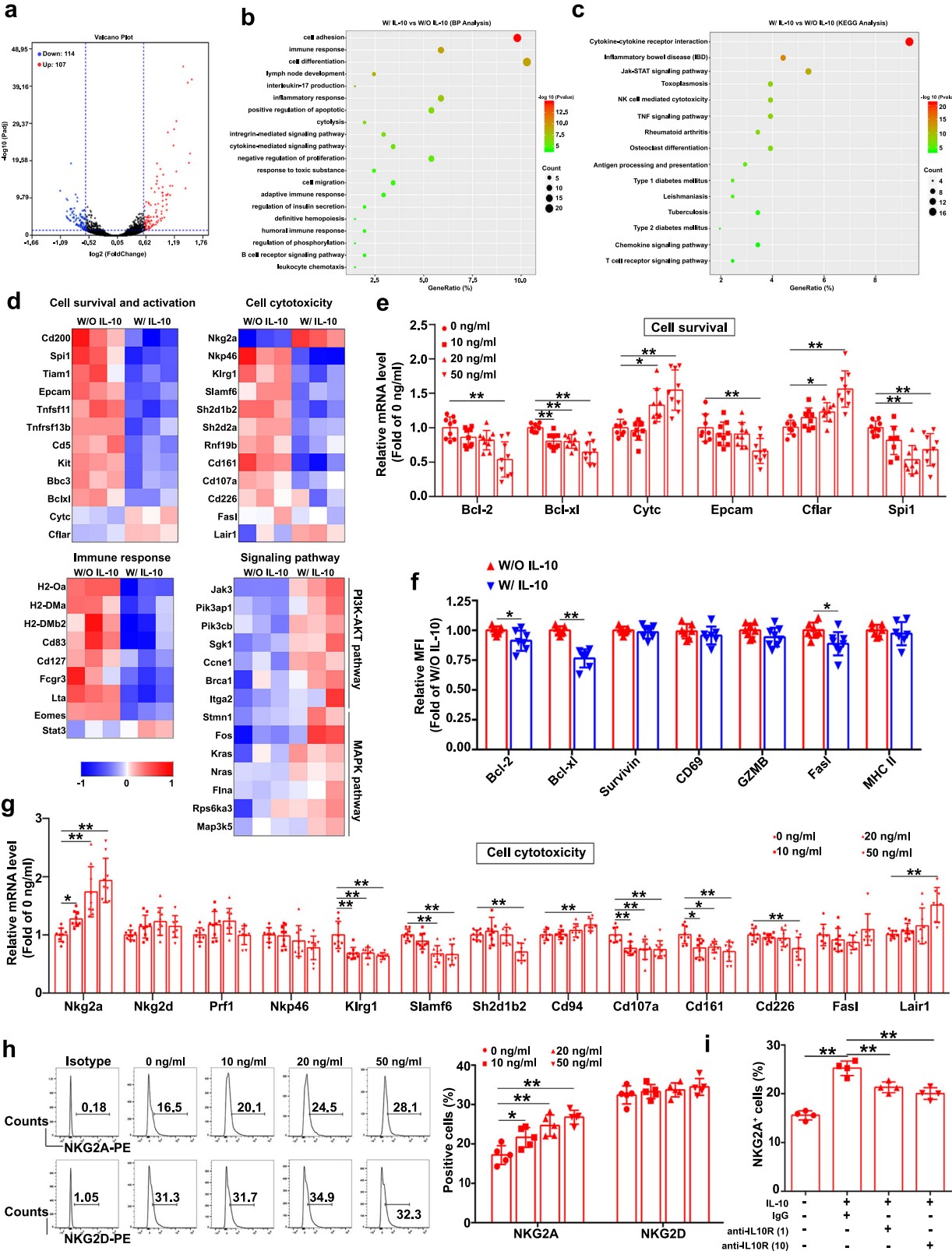

regulating cDNT activation and function need to be further studied.

In this study, we mainly found cDNT selectively suppressed M1 macrophages, but not M2 macrophages during NASH development. Recent studies underlie the new possibilities and more accurate differentiation of hepatic macrophages by using scRNA-seq[45–47], thus scRNA-seq study might be more helpful to

further reveal the macrophage population and polarization after cDNT transfer, we will perform the study in the future.

In summary, as shown in Fig. 8, the administration of cDNT could prevent the development and progression of obesity, insulin resistance, and NASH. The cDNT not only decreased the CD4+ T cell proportion and survival, but also selectively suppressed the proportion of liver-infiltrating M1 macrophages and the secretion

**Fig. 5 Transcriptome sequencing analysis of IL-10-mediated regulation of cDNT. a** The distributions of upregulated (107) and downregulated genes (114; ≥1.5-fold difference, $p$adj < 0.05) in cDNT treated with or without IL-10 are depicted in a volcano plot. **b, c** Biological process (BP) and KEGG pathway analyses were performed based on the significantly upregulated and downregulated genes in cDNT. **d** Heatmap showing the upregulated and downregulated genes related to cell survival and activation, cell cytotoxicity, immune responses, and signaling pathways in cDNT. **e** The significantly changed genes related to cell survival were confirmed via real-time PCR after IL-10 stimulation of cDNT ($n = 9$ biologically independent samples per group). Actual $p$ values (left → right): 0.00026, 0.0037, 0.0028, 6.50e−7, 0.012, 0.000024, 0.0017, 0.048, 5.68e−7, 0.000081, 0.0072. **f** Relative expression of cell markers plotted as the fold change in expression compared to cDNT without IL-10 stimulation ($n = 6$ biologically independent samples per group). W/O IL-10, red; W/IL-10, blue. Actual $p$ values (left → right): 0.027, 0.000008, 0.030. **g** The expression of cell cytotoxicity-related genes were determined via real-time PCR ($n = 9$ biologically independent samples per group). Actual $p$ values (left → right): 0.047, 0.000076, 0.000002, 0.00019, 0000025, 00000024, 0.00015, 0.000097, 0.0031, 0.000069, 0.0078, 0.0046, 0.0038, 0.012, 0.020, 0.0011, 0.0054, 0.00022. **h** Representative flow cytometry plots (left) and statistical analysis (right) of NKG2A$^+$ and NKG2D$^+$ cells among cDNT after IL-10 stimulation ($n = 5$ biologically independent samples per group). Actual $p$ values (left → right): 0.038, 0.00067, 0.000045. **i** cDNT incubated with anti-CD3/CD28 antibodies were pretreated with anti-IL-10R neutralizing mAb (1 and 10 μg/ml) or isotype control for 4 h, and cocultured with IL-10 (50 ng/ml) for 24 h. Statistical analysis of NKG2A$^+$ cells relative to cDNT in each group ($n = 4$ biologically independent samples per group). Actual $p$ values (left → right): 5.06e−7, 0.0030, 0.00028. Data are presented as the mean ± SD. Statistical analysis for **f** was performed by Student's $t$ test, and others were performed by one-way ANOVA with post hoc multiple comparisons test. Two-sided $p$ values < 0.05 were considered significant. *$p$ < 0.05; **$p$ < 0.01. Source data, including exact $p$ values, are provided as a Source data file.

of proinflammatory cytokines. Compared with M1 macrophages, M2 macrophages were less susceptible to cDNT-mediated cytotoxicity, which might be due to the downregulation of cDNT immunosuppressive function by IL-10 secreted by M2 macrophages. Furthermore, we found that IL-10 could inhibit cDNT survival and promote expression of the inhibitory molecule NKG2A, which contribute to the decreased suppressive function of cDNT.

In conclusion, ex vivo-generated CD4$^+$ T cell cDNT exert potent protection against diet-induced obesity, type 2 diabetes, and NASH. The improved outcomes are due to the inhibition of liver inflammatory cells and the restoration of liver immune homeostasis. This study supports the concept and feasibility of this autologous immune cell-based therapy for the treatment of NASH.

## Methods

**Animals**. Eight-week-old weight-matched male C57BL/6 mice and CD45.1 congenic C57BL/6 mice were purchased from The Jackson Laboratory (ME, USA). The C57BL/6 mice were fed either a normal control diet (NCD), a HFD (D12492, Research Diets, NJ, USA), a MCD (10401, Beijing HFK Bioscience, Beijing, China), or a CD-HFD (Research Diets, D05010402). C57BL/6 mice fed with MCD and treated with rapamycin (0.5 mg/kg/day, intraperitoneally) as a positive control for 4 weeks. The mice were maintained in a pathogen-free, temperature-controlled environment with a 12-h light/dark cycle. The animal studies were performed in compliance with the ethical guidelines for animal studies and approved by the Institutional Animal Care and Ethics Committee of Beijing Friendship Hospital.

**Conversion of DNT in vitro**. The conversion of CD4$^+$ T cells to cDNT was described previously[21]. Briefly, CD4$^+$CD25$^-$ T cells were isolated from CD45.1-positive C57BL/6 mice spleens and LN. Then, the purified CD4$^+$CD25$^-$ T cells were cocultured with C57BL/6 mature DCs and rmIL-2 (50 ng/ml) for 7 days. cDNT were sorted from culture through a fluorescence-activated cell sorter (FACS Aria II, BD Biosciences).

**cDNT adoptive transfer and anti-IL-10 neutralizing antibody administration in vivo**. In the HFD-fed NASH model, after 8 weeks of HFD feeding, C57BL/6 mice received an adoptive transfer of $5 \times 10^6$ CD45.1-positive cDNT (purity > 97%) and then were continuously fed the HFD for another 8 weeks. In the MCD-fed NASH model, the C57BL/6 mice received an adoptive transfer of $5 \times 10^6$ CD45.1-positive cDNT (purity > 97%) and then were fed the MCD for 4 weeks. In the CD-HFD-fed NASH model, C57BL/6 mice were fed a CD-HFD for 12 weeks, received a transfer of $5 \times 10^6$ CD45.1-positive cDNT (purity > 97%), and then continued to be fed the CD-HFD for another 4 weeks. For anti-IL-10 neutralizing antibody treatment model, MCD-fed mice were treated by intraperitoneal injection with 0.3 mg of anti-IL-10 Ab (clone JES5-2A5, BioXCell, West Lebanon, NH, USA) every other day after cDNT transfer for 4 weeks, then liver tissue inflammation and lymphocytes proportion were detected.

**Intraperitoneal glucose tolerance test and insulin tolerance test**. Glucose tolerance tests (GTTs) and insulin tolerance tests (ITTs) were performed after a 6-h fast. For GTTs, mice were injected intraperitoneally with 1 g/kg glucose. For ITTs, mice were injected with 0.75 units/kg regular human insulin (Humulin R, Lilly,

USA). Mouse blood was collected at 0, 30, 60, 90, 120, and 150 min after insulin or glucose injection. Plasma glucose concentration was measured with a Medisafe blood glucose meter (Terumo Corporation, Japan).

**Flow cytometry analysis**. The isolation and examination of hepatic immune cells were described previously[8]. Immune cells were harvested and analyzed to determine the expression levels of various cell surface and intracellular markers. All samples were acquired on a FACS Aria II flow cytometer (BD Biosciences), and data were analyzed using FlowJo software (Tree Star, OR, USA).

**Real-time PCR**. Total RNA was isolated from liver tissue, adipose tissue, or cDNT using a RNeasy Plus Mini kit (QIAGEN, Germany), and reverse transcribed into cDNA using a SuperScript III RT kit (Invitrogen, CA, USA). Real-time PCR was performed by the ABI 7500 sequence detection system (Applied Biosystems, CA, USA). Amplicon expression in each sample was normalized to GAPDH expression, and gene expression was subsequently quantified using the $2^{-\Delta\Delta Ct}$ method. The genes and primer sequences are shown in Table S1.

**Histology and immunohistochemistry**. Mouse liver tissues were fixed, dehydrated, embedded in paraffin, and cut into 6-μm sections. Hematoxylin and eosin (H&E) staining, oil red o staining, and Sirius red staining were performed by Servicebio Technology (Wuhan, China). H&E-stained liver sections were assessed by three experienced liver pathologists for an NAS (based on steatosis, inflammation, and ballooning). For immunohistochemistry, pAb against α-SMA (Servicebio) was used to detect activated hepatic stellate cells. Liver-infiltrating monocytes, T cells, and B cells were evidenced in paraffin sections using anti-mouse CD45, CD3, CD11b, CD20, and F4/80 antibody (Abcam, Cambridge, MA, USA). Immunopositive cells were counted by ImageJ software (National Institutes of Health, USA).

**Western blot**. Liver tissues were solubilized in RIPA (Solarbio, Beijing, China) and the protein concentration was then determined using BCA protein assay (Thermo Fisher Scientific, Inc., Pittsburgh, PA, USA). Protein mixtures normalized to 100 μg separated by 12% SDS–PAGE and transferred onto PVDF membrane (Millipore, Bedford, MA, USA). Primary antibodies against α-SMA (diluted 1:2000, Abcam), TNF-α (diluted 1:500, Abcam), IL-10 (diluted 1:500, Abcam), and β-actin (diluted 1:5000, Abcam) were used. The membranes were incubated with secondary antibodies conjugated to HRP, and the proteins were detected by enhanced chemiluminescence (Thermo). The relative density of the protein bands was quantitatively determined using ImageJ software.

**Induction of M1 and M2 macrophages**. Bone marrow was collected from CD45.1-positive C57BL/6 mice, erythrocytes were lysed, and bone marrow cells were resuspended into a single-cell suspension. Gr1$^+$, B220$^+$ and Ter119$^+$ cells were removed by magnetic column selection, and the remaining cells were monocytes. These monocytes were stimulated with 20 ng/ml GM-CSF for 5 days in six-well plates ($2 \times 10^6$ cells per well). After 5 days, M1 macrophages were induced by 100 ng/ml LPS and 20 ng/ml IFN-γ, and M2 macrophages were induced by 20 ng/ml IL-4 for 24 h.

**Transcriptome sequencing analysis**. The cDNT stimulated with anti-CD3/CD28 antibodies and IL-10 (50 ng/ml) for 24 h, then total RNA was isolated. Transcriptome sequencing libraries were generated using NEBNext® Ultra™ RNA Library Prep Kit for Illumina® (NEB, USA), following manufacturer's recommendations and sequenced on an Illumina Hiseq platform (Illumina, San Diego, CA). Sequences were aligned to the reference genome with TopHat and processed

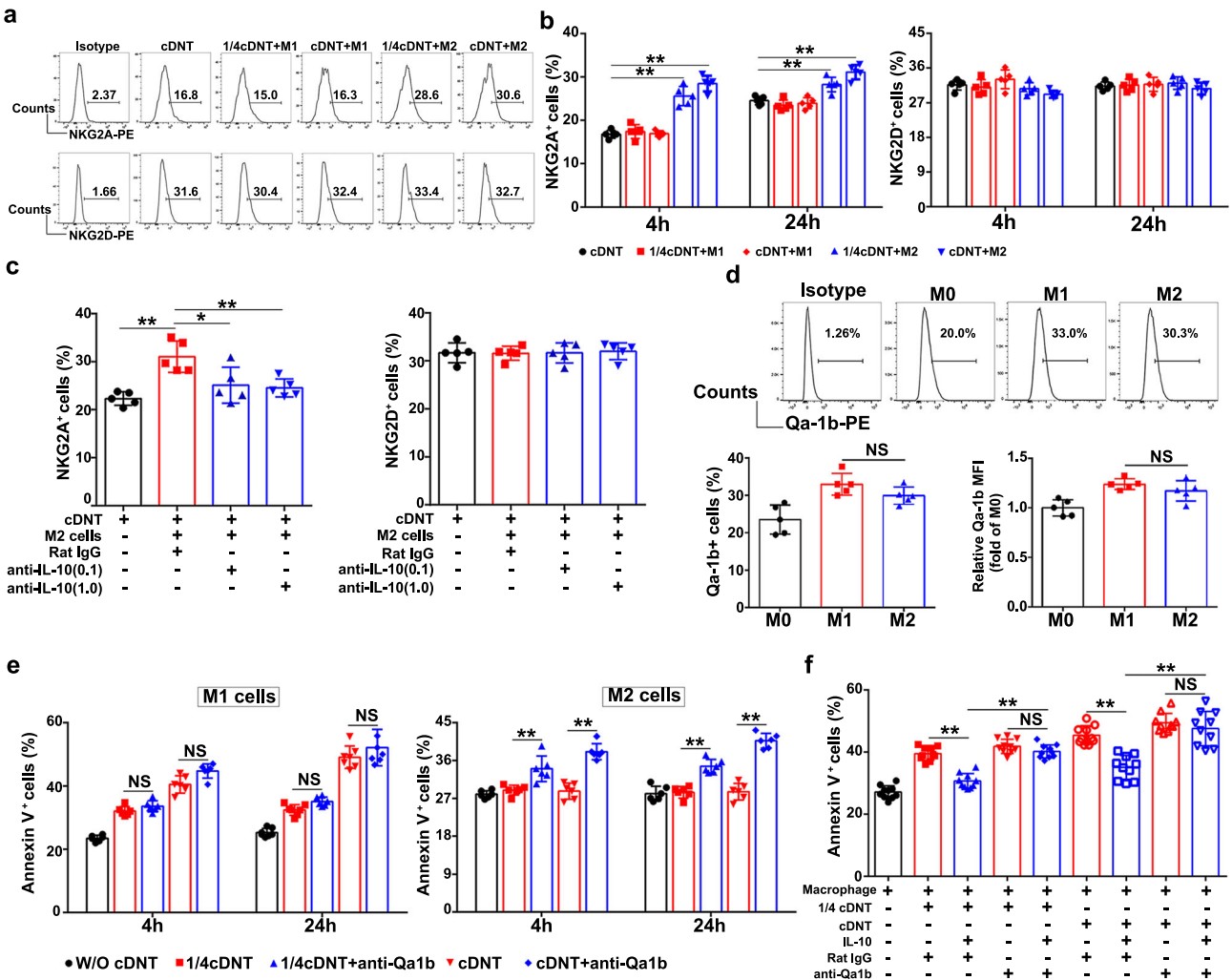

**Fig. 6 IL-10 secreted by M2 macrophages decreased the suppressive function of cDNT by increasing the NKG2A/Qa-1b interaction. a, b** Representative flow cytometry plots (**a**) and statistical analysis (**b**) of NKG2A+ and NKG2D+ cDNT after coculture with M1/M2 macrophages for 4 and 24 h ($n = 5$ biologically independent samples per group). cDNT, black; cDNT + M1, red; cDNT + M2, blue. Actual $p$ values (left → right): 1.67e−7, 1.34e−9, 0.0034, 0.00003. **c** M2 macrophages were pretreated with anti-IL-10 neutralizing mAb (0.1 and 1.0 μg/ml) or IgG control for 4 h and coculture with cDNT for 24 h. Statistical analysis of NKG2A+ and NKG2D+ cDNT in each group ($n = 5$ biologically independent samples per group). cDNT, black; cDNT + M2, red; cDNT + M2 + anti-IL-10 mAb, blue. Actual $p$ values (left → right): 0.00067, 0.017, 0.0090. **d** Flow cytometric images (top) and statistical analysis (bottom) of Qa-1b+ cells among different types of macrophages ($n = 5$ biologically independent samples per group). M0, black; M1, red; M2, blue. Actual $p$ values (left → right): 0.30, 0.42. **e** M1 and M2 macrophages were pretreated with anti-Qa-1b neutralizing mAb (10 μg/ml) or IgG control for 4 h before cocultured with cDNT. Statistical analysis of Annexin V+ cells relative to M1 and M2 macrophages in each group ($n = 6$ biologically independent samples per group). W/O cDNT, black; W/ cDNT, red; W/cDNT+anti-Qa-1b, blue. Actual $p$ values (left → right): for M1 cells, 0.64, 0.059, 0.58, 0.48; for M2 cells, 0.00096, 5.34e−8, 0.000014, 4.52e−11. **f** Macrophages were pretreated with anti-Qa-1b neutralizing mAb (10 μg/ml) or control IgG for 4 h and then cocultured with cDNT pretreated with or without IL-10 (50 ng/ml) for 24 h. Statistical analysis of Annexin V+ cells relative to macrophages in each group ($n = 10$ biologically independent samples per group). Actual $p$ values (left → right): 4.50e−7, 5.34e−8, 0.95, 3.42e−9, 3.66e−12, 0.93. Data are presented as the mean ± SD. Statistical analysis was performed by one-way ANOVA with post hoc multiple comparisons test. Two-sided $p$ values < 0.05 were considered significant. *$p$ < 0.05; **$p$ < 0.01; NS not significant. Source data, including exact $p$ values, are provided as a Source data file.

with Cufflinks, which quantifies each transcript in each sample using reference annotations produced by the University of California Santa Cruz UCSC. Differentially expressed genes with a fold change of ≥1.5 and $p$adj < 0.05 between with IL-10 or without IL-10-treated cDNT were submitted to GO and KEGG enrichment analysis, which uses unbiased methods to assess pathway enrichment. The mRNA sequencing data described in this study were uploaded to the National Center for Biotechnology Information (NCBI) Gene Expression Omnibus (accession no. GSE134346).

**In vitro suppression assays**. The suppression assays of cDNT on Th17 cells were performed as follows. Naive CD45.1-positive CD4+ T cells (3 × 10^5/well) were isolated from mice spleens, and stimulated with anti-CD3 (2 μg/ml), anti-CD28 (1 μg/ml), IL-6 (2 ng/ml), TGF-β (1 ng/ml), anti-IL-4 (2 μg/ml), and IFN-γ

(5 μg/ml) in 96-well flat-bottom culture plates for 3 days for Th17 cells differentiation. After 3 days culture, 0.75 × 10^5/well and 3 × 10^5/well converted CD45.2+ cDNT were added into the culture for 24 h. Then Annexin V staining and IL-17 intracellular staining were performed to detect the apoptosis and Th17 differentiation of CD45.1+ CD4+ T cells.

The suppression assays of cDNT to macrophages were performed as follows. CD45.1-positive M1 and M2 macrophages (2 × 10^5 cells/well) were cocultured with cDNT at a ratio of 4:1, 2:1, or 1:1 in 96-well round-bottom culture plates. After 4 or 24 h, Annexin V, TNF-α, NKG2A, and NKG2D staining were performed to detect protein expression levels and the apoptosis of CD45.1-positive cells via flow cytometry.

**Statistical analysis**. Statistical analysis was performed using SPSS Statistics (IBM SPSS Statistics for Windows, Version 22.0. Armonk, NY, USA) and Prism

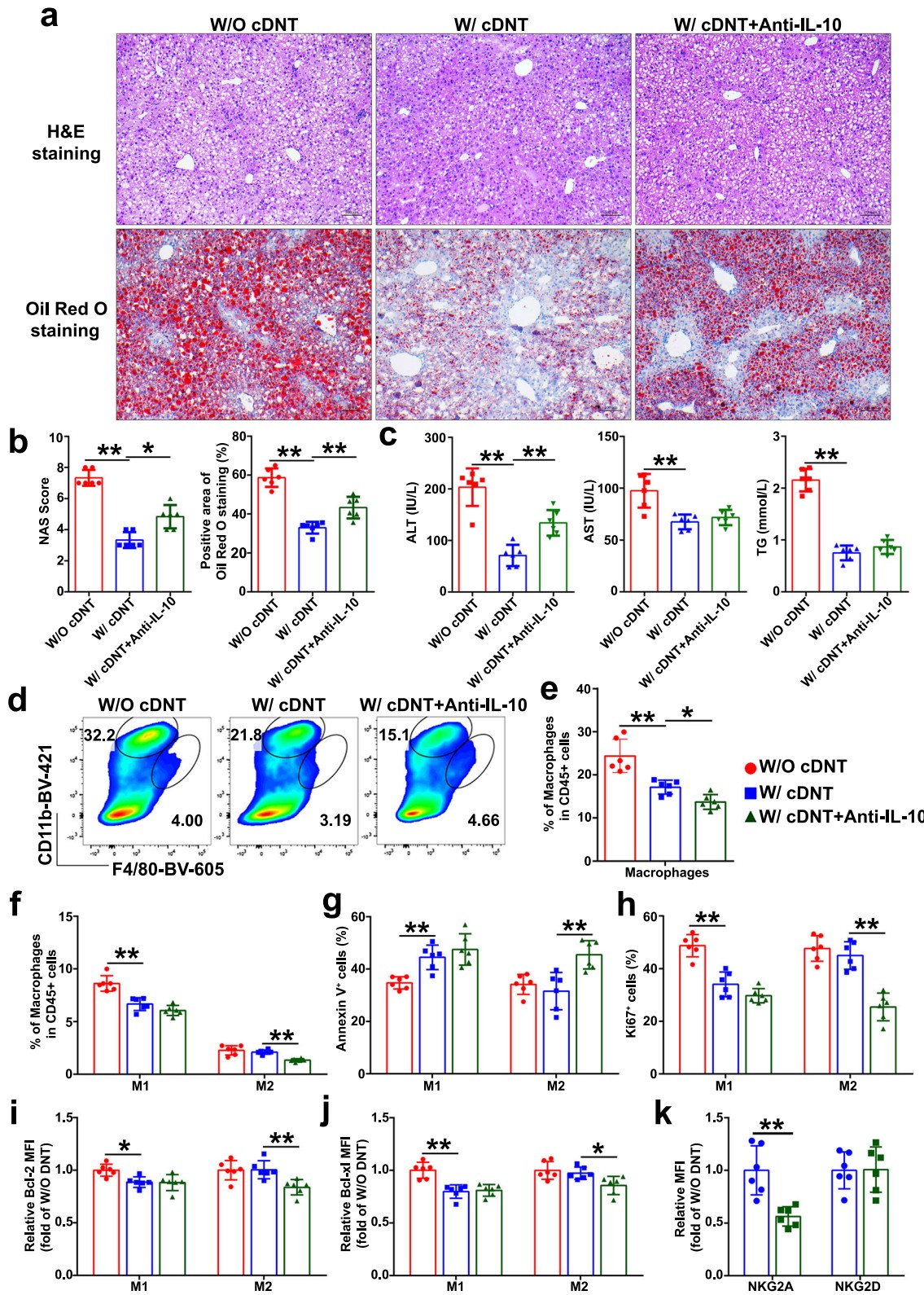

6.0 software (GraphPad Software, San Diego, CA, USA). Values are expressed as the mean ± standard deviation (SD). The normal distribution of variables was tested with the Shapiro–Wilk test. Differences between two groups were compared by $t$ test for normal variables and Kruskal–Wallis test for non-normal variables. One-way ANOVA with post hoc test for normal variables and Kruskal–Wallis test for non-normal variables were used in multiple comparisons. Two-sided $p$ values <0.05 were considered significant.

**Reporting summary**. Further information on research design is available in the Nature Research Reporting Summary linked to this article.

## Data availability

RNAseq data from cDNT cells with or without IL-10 stimulation generated in this study, have been deposited in the GEO database under accession number GSE134346. All the

**Fig. 7 Anti-IL-10 neutralizing antibody increased cDNT cytotoxicity toward M2 macrophages in vivo. a** Representative H&E staining and oil red o staining in liver paraffin sections of mice fed with MCD diet for 4 weeks. Scale bars, 100 μm ($n = 6$ biologically independent samples per group). **b** Statistical analysis of NAS score and positive area of oil red o staining in each group. W/O cDNT, red; W/ cDNT, blue; W/ cDNT+anti-IL-10, green. Actual $p$ values (left → right): 0.0027, 0.022, 2.03e−7, 0.0037. **c** Plasma ALT, AST, and TG levels were measured. Actual $p$ values (left → right): 0.000002, 0.0039, 0.00077, 6.81e−9. **d** Representative flow cytometry plots of hepatic macrophages in MCD-fed mice. **e, f** Statistical analysis of the percentages of macrophages (**e**), M1 and M2 macrophages (**f**) among CD45.2+ cells in liver tissue. Actual $p$ values (left → right): **e** 0.00042, 0.048; **f** 0.000017, 0.0011. **g, h** The Annexin V (**g**) and Ki67 staining (**h**) of hepatic M1 and M2 macrophages in MCD-fed mice with or without anti-IL-10 administration for 4 weeks. Actual $p$ values (left → right): **g** 0.0060, 0.0018; **h** 0.000032, 0.000023. **i, j** Bcl-2 and Bcl-xl expression levels in intrahepatic M1 and M2 macrophages determined by flow cytometry, and relative changes of MFI from each group were determined. Actual $p$ values (left → right): **i** 0.017, 0.0095; **j** 0.00023, 0.049. **k** Relative changes of NKG2A and NKG2D expression in CD45.1+ cDNT in MCD-fed mice with or without anti-IL-10 treatment. **p = 0.002$. Data are presented as the mean ± SD; $n = 6$ mice/group. Statistical analysis for the first left figure in **b** (NAS score group) was performed by Kruskal–Wallis multiple comparisons test, statistical analysis for **k** was performed by Student's $t$ test, and others were performed by one-way ANOVA with post hoc multiple comparisons test. Two-sided $p$ values < 0.05 were considered significant. *$p < 0.05$; **$p < 0.01$. Source data, including exact $p$ values, are provided as a Source data file.

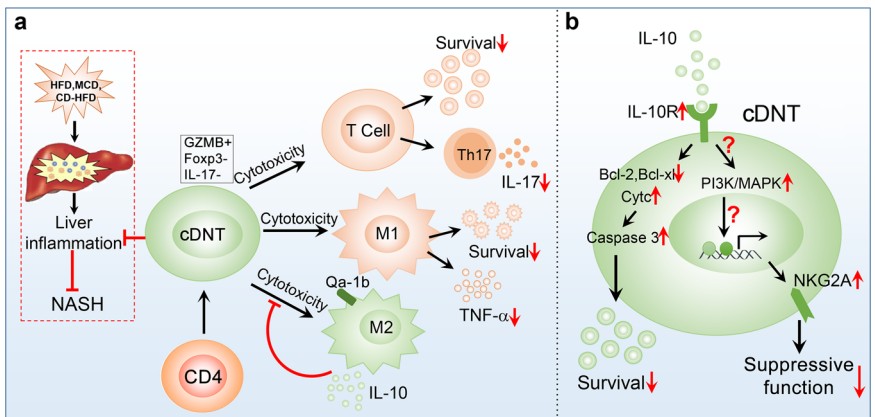

**Fig. 8 Mechanisms by which transferred cDNT regulate the liver immune microenvironment and inhibit liver inflammation during NASH development. a** The transferred cDNT not only decreased CD4+ T cell proportion and survival and Th17 cell differentiation, but also selectively suppressed the proportion and cytokine production of liver-infiltrating M1, but not M2 macrophages. IL-10 secreted by M2 macrophages downregulated the immunosuppressive function of cDNT toward M2 macrophages. **b** The IL-10/IL-10R interaction could decrease Bcl-2 and Bcl-xL expression, and enhance cytochrome C leakage and caspase-3 activity, leading to decreased cDNT survival. The IL-10/IL-10R interaction also activates the PI3/MAPK signaling pathway, which might induce the expression of the inhibitory receptor NKG2A. The inhibition of cDNT survival and the promotion of NKG2A expression by the IL-10/IL-10R interaction contribute to decreasing cDNT function.

other data supporting the findings of this study are available within the article and its Supplementary Information files, and from the corresponding author upon reasonable request. Source data are provided with this paper.

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

## Acknowledgements

This work was supported by grants from the National Natural Science Foundation of China (81870399 and 81970503) and the Beijing Natural Science Foundation (7192046 and 7172060).

## Author contributions

All listed authors participated meaningfully in the study, and that they have seen and approved the submission of this manuscript. G.S. participated in performing the research, analyzing the data, and initiating the original draft of the article. X.Z., M.L., C.Z., H.J., L.L., C.L., Y.W., W.S., D.T., H.X., Y.T., Y.W., and K.L. participated in performing the research and collecting the data. D.Z. and Z.Z. established the hypotheses, supervised the studies, analyzed the data, and cowrote the manuscript.

## Competing interests

G.S., X.Z., W.S., D.T., H.X., Y.T., K.L., and D.Z. are inventors of a Chinese patent for the method of ex vivo conversion of DNT. The remaining authors declare no conflict of interest.
