## [Peer Review File · Nature Communications]

Reviewers' Comments:

Reviewer #1:

Remarks to the Author:

NCOMMS-19-32808

Double negative T-cells prevent the development and progression of nonalcoholic steatohepatitis.

NAFLD and its progressive form NASH are among the top causes of cirrhosis and hepatocellular carcinoma and the leading indications for liver transplantation. However, there are currently no approved treatments for NASH/NAFLD. In this study, we adoptively transferred ex vivo-generated double-negative T cells (DNT) to mice with diet-induced NASH. The DNT-treated mice exhibited significantly less weight gain, improved glucose tolerance and insulin sensitivity, decreased liver fat accumulation, lobular inflammation and focal liver necrosis. DNT selectively suppressed liver-infiltrating Th17 cells and proinflammatory M1, not immune regulatory M2, macrophages. IL-10 secreted by M2 macrophages decreased the survival and function of DNT to protect M2 macrophages from DNT-mediated lysis. NKG2A, a cell inhibitory molecule, was upregulated by IL-10, contributed to increased apoptosis, and dampened the suppressive function of DNT. This study supports the concept and the feasibility of potentially utilizing this novel autologous immune cell-based therapy for the treatment of NASH.

The findings of this paper are rather surprising given that DN T cells are generally considered a pro-inflammatory subset within the T cell compartment both in men and mice (1-4). DN T cells exhibit greatly elevated metabolic activity, which is characterized by accumulation of oxidative stress generating mitochondria and activation of the mechanistic target of rapamycin (5). The expansion of DN T cells has been widely linked to pathogenesis of lupus both in mice and humans (5-11). Importantly, lupus patients and mice also exhibit oxidative stress-driven inflammation of the liver which benefits from depletion of DN T cells by using antioxidants, such as acetylcysteine or rapamycin (12,13). DN T cells signal through the TCR and PD1 which have not been evaluated or even pictured in the proposed mechanism of action in Figure 7. Likewise, mTOR activation is critical for DN T cell function and T-cell lineage specification in general, which is also missing from the authors' concept. Importantly, rapamycin also blocks diet-induced NALFD (Hepatology. 2014 Nov;60(5):1581-92). Therefore, it would be important to evaluate TCR signaling and use rapamycin as a positive control for the clinical efficacy of DN T cell therapy in steatohepatitis.

Specific comments:

Figure 1. This figure supports the contention that a single transfer of DN T cells prevented diet-induced obesity, insulin resistance and NASH development. Apparently, mice that received DNT therapy had lower plasma IFN- γ , TNF- α , IL-6, IL-9, IL-17A and IL-17F levels (Figure 1K). Fibrosis-related genes, such as SMA, Col1a1 and Col3a1, were also downregulated in the livers of HFD-fed mice that underwent DNT transfer (Figure 1L). This figure should evaluate the functionality of DN T cells and their ability to produce pro-inflammatory cytokines.

Figure 2. T cell subsets, including CD4, CD8, DN, and Treg cells and NK cell subsets and their intracellular cytokine production profiles should be presented as flow cytometry dot plots including the gating strategy. It would be important to systematically present cell populations from liver, spleen, and lymph node of the same mice.

Abstract: The statement "there are currently no approved treatments for NASH/NAFLD" is too vague and it should be removed from the abstract. One might consider replacement with effective treatment. However, this statement rather belongs to the discussion clarifying what "approved"

means. Again, it would be important to use rapamycin as a positive controls which eliminates proinflammatory T cell development.

Reference List

1. Crispin, J. C., and G. C. Tsokos. 2009. Human TCR-alpha beta+ CD4- CD8- T cells can derive from CD8+ T cells and display an inflammatory effector phenotype. *J. Immunol.* 183:4675-4681.
2. Rodriguez-Rodriguez, N., S. A. Apostolidis, L. Fitzgerald, B. S. Meehan, A. J. Corbett, J. M. Martin-Villa, J. McCluskey, G. C. Tsokos, and J. C. Crispin. 2016. Pro-inflammatory self-reactive T-cells are found within murine TCR-ab+CD4-CD8-PD-1+ cells. *Eur J Immunol* 46:1383-1391.
3. Apostolidis, S. A., T. Rauen, C. M. Hedrich, G. C. Tsokos, and J. C. Crispin. 2013. Protein phosphatase 2A enables expression of interleukin 17 (IL-17) through chromatin remodeling. *J. Biol. Chem.* 288:26775-26784.
4. Tsokos, G. C. 2011. Systemic Lupus Erythematosus. *N. Engl. J. Med.* 365:2110-2121.
5. Lai, Z.-W., R. Hanczko, E. Bonilla, T. N. Caza, B. Clair, A. Bartos, G. Miklossy, J. Jimah, E. Doherty, H. Tily, L. Francis, R. Garcia, M. Dawood, J. Yu, I. Ramos, I. Coman, S. V. Faraone, P. E. Phillips, and A. Perl. 2012. N-acetylcysteine reduces disease activity by blocking mTOR in T cells of lupus patients. *Arthritis Rheum.* 64:2937-2946.
6. Crispin, J. C., M. Oukka, G. Bayliss, R. A. Cohen, C. A. Van Beek, I. E. Stillman, V. C. Kytтары, Y. T. Juang, and G. C. Tsokos. 2008. Expanded Double Negative T Cells in Patients with Systemic Lupus Erythematosus Produce IL-17 and Infiltrate the Kidneys. *J. Immunol.* 181:8761-8766.
7. Hedrich, C. M., T. Rauen, J. C. Crispin, T. Koga, C. Ioannidis, M. Zajdel, V. C. Kytтары, and G. C. Tsokos. 2013. cAMP-responsive Element Modulator alpha (CREM-alpha) trans-Represses the Transmembrane Glycoprotein CD8 and Contributes to the Generation of CD3+CD4-CD8- T Cells in Health and Disease. *J. Biol. Chem.* 288:31880-31887.
8. Koga, T., C. M. Hedrich, M. Mizui, N. Yoshida, K. Otomo, L. A. Lieberman, T. Rauen, J. C. Crispin, and G. C. Tsokos. 2014. CaMK4-dependent activation of AKT/mTOR and CREM-a underlies autoimmunity-associated Th17 imbalance. *J. Clin. Invest.* 124:2234-2245.
9. Apostolidis, S. A., N. Rodriguez-Rodriguez, A. Suarez-Fueyo, N. Dioufa, E. Ozcan, J. C. Crispin, M. G. Tsokos, and G. C. Tsokos. 2016. Phosphatase PP2A is requisite for the function of regulatory T cells. *Nature Immunology* 17:556.
10. Lai, Z.-W., R. Borsuk, A. Shadakshari, J. Yu, M. Dawood, R. Garcia, L. Francis, H. Tily, A. Bartos, S. V. Faraone, P. E. Phillips, and A. Perl. 2013. mTOR activation triggers IL-4 production and necrotic death of double-negative T cells in patients with systemic lupus erythematosus. *J. Immunol.* 191:2236-2246.
11. Lai, Z., R. Kelly, T. Winans, I. Marchena, A. Shadakshari, J. Yu, M. Dawood, R. Garcia, H. Tily, L. Francis, S. V. Faraone, P. E. Phillips, and A. Perl. 2018. Sirolimus in patients with clinically active systemic lupus erythematosus resistant to, or intolerant of, conventional medications: a single-arm, open-label, phase 1/2 trial. *Lancet* 391:1186-1196.
12. Liu, Y., J. Yu, Z. Oaks, I. Marchena-Mendez, L. Francis, E. Bonilla, P. Aleksiejuk, J. Patel, K. Banki, S. K. Landas, and A. Perl. 2015. Liver injury correlates with biomarkers of autoimmunity and disease activity and represents an organ system involvement in patients with systemic lupus erythematosus. *Clin. Immunol.* 160:319-327.
13. Oaks, Z., T. Winans, T. Caza, D. Fernandez, Y. Liu, S. K. Landas, K. Banki, and A. Perl. 2016. Mitochondrial dysfunction in the liver and antiphospholipid antibody production precede disease onset and respond to rapamycin in lupus-prone mice. *Arthritis Rheumatol.* 68:2728-2739.

Reviewer #2:

Remarks to the Author:

The manuscript by Sun et al. addresses the potential of adoptively transferred double-negative T cells to prevent the development of NASH in a mouse model based on feeding a high-fat diet. Moreover, they describe a potential mechanism regulating the suppressive activity of these cells,

which seems to be based on interaction between the inhibitory receptor NKG2A and Qa-1.

The study is of potential interest notably in the mechanistic part; however, it is rather weak in its in vivo part.

Specific comments:

1. The NASH model is problematic. Although I can appreciate the effect of the transferred DN T cells on steatosis and metabolic parameters, it is hard to see the effect on inflammation. Mainly, because the degree of inflammation and liver damage in the high-fat diet group not receiving DN T cells is very mild. The ALT/AST levels are only slightly increased. The HE staining does not show significant infiltration. In other words: there is almost no hepatitis. Therefore, I find it difficult to follow the authors' claim that DN T cells could be used for NASH treatment.

(By the way, the authors should take care that the HE sections that are compared between the groups are similar with respect to size and distribution of portal fields and central veins; now they compare sections showing only veins with sections showing mostly portal tracts.)

2. Along the same line: if the authors wish to claim an treatment effect on fibrosis, they should show it not merely by PCR for some markers, but by more adequate assay, such as Sirius red staining, hydroxyproline content, aSMA immunostaining or the like.

3. The time-point of CD45.1 analysis after transfer of DN T cells (Fig 1C) is not clear. Actually, one would like to see several time-points to get an idea of the dynamics of survival and tissue distribution.

4. The method of DN T cell generation is not described. Mere reference to a previous paper is insufficient (Ref.16), as there are several alternative methods described. The details of the applied cytokines, APCs, number and chronology of re-stimulations, and the like are essential, but lacking.

5. On page 9 in the context of Figure 2, the authors conclude 'that a single transfer of DNT to HFD-fed mice could reduce hepatic inflammation and NASH development by decreasing hepatic CD4+ T-cell proportion and survival and Th17 cell differentiation'. On the basis of the current data, I think that this is merely an association. To really sustain their statement, it would require some kind of inhibition experiment, which is however not provided.

6. The in vitro experiments showing the possible function of IL-10, NKG2A, and Qa-1 are interesting. However, to really claim a clear functional relevance, it will be necessary to perform inhibition experiments in vivo, e.g. using inhibitory antibodies to IL-10 or IL10R, or Qa-1. However, I am afraid that such experiments will again be limited by the inherent problems of the NASH model used by the authors.

7. Minor points are related to suboptimal gating strategy for DCs as MHC II/CD11c high (what about pDCs?), and M1/M2 based merely on CD206 and MHC II.

8. I have some doubt that the use of the t test is really appropriate, as I do not think that it normally distributed.

Reviewer #3:

Remarks to the Author:

In their present study, the authors aimed to find a potential new therapy for NASH by allograft transfer of double negative T-cells (DNT) in steatohepatitis mouse models. By transfer of DNT into mice with NASH generated by different dietary models they found that DNT ameliorates inflammation and consequently improved NASH and glucose tolerance.

The main target of DNT was a reduction in inflammatory M1 macrophages. Subsequent cell culture

analysis suggest that M2 macrophages are protective by IL-10 secretion via NKG2A/Qa1b signaling.

To address their query the authors used a well-structured approach and study design. Background information on their main methods was presented and results and evidence were conclusively constituted. Different NASH-models and cell culture stimulation approaches as well as cohesive methods like mRNA levels and protein expression were used to underline their findings.

Although the results are convincing, some major and minor points should be addressed to underline the study outcome.

Major points:

1. Liver histology should be further investigated and quantified. Specifically severity of NASH has to be evaluated by a blinded and experienced pathologist, e.g. using the international approved NAS score. Moreover, fibrosis must be measured by Sirius Red staining and lipomatosis should also be addressed in the HFD-fed mice by Oil Red O staining. All staining should be quantified. Differences in fibrosis related protein expression should be also affirmed by western blot analysis.
2. The immune phenotype must be confirmed in liver histology by different immune staining methods. First of all the number of infiltrating leukocytes should be validated and quantified by CD45 staining (IHC and/or IF). Subpopulations could be addressed by CD3, CD20, CD11b and F4/80 stainings. Moreover, single-cell RNA sequencing (scRNA-seq) should be included to identify and differentiate phenotypes of M1 and M2 macrophages since the identification just by FACS markers can be improved.
3. The same histological immune quantification should be done for VAT. Furthermore, size of adipocytes has to be quantified to show significant differences.
4. The authors describe in Figure 1C that DNT accumulate in Liver and VAT. In supplemental Figure 1A VAT is not shown but here draining lymph nodes are stated as major accumulation points. Can the authors explain this difference between those models? Why are lymph nodes not shown for HFD-fed mice? How do the authors clarify that it is only draining lymph nodes of the liver? Can they show data for non-draining lymph nodes and explain how they identified draining lymph nodes? Do they also have data of VAT in this model?
5. Protein levels are shown by FACS analysis. The most important proteins in this study (e.g. IL-10, TNF α) should also be measured by western blot analysis to verify different protein levels.
6. In their abstract and discussion, authors state that among lymphocytes specifically TH17 cells are suppressed by DNT and confirm this in the animal models by FACS and mRNA levels. Can they explain or discuss the cause of this TH17-specific effect? Can they also contribute cell culture models to ensure this effect on TH17 cells?
7. The observed effect in vitro should be confirmed by in vivo experiments as well. Are macrophages and DNT in vivo affected when animals are treated with anti-IL-10 neutralizing mAb?

Minor points:

1. Authors discuss that the accumulation of DNT in liver and VAT could be due to CXCR3-CXCL10 interaction. Authors could measure CXCL10 levels in liver and VAT to address this point. It could also be interesting to look at possible differences between DNT and not DNT treated animals.
2. Since more than half of the results refer to DNT and macrophage interaction, this topic should be mentioned in the title, e.g. "Double negative T-cells prevent the development and progression of nonalcoholic steatohepatitis by suppression of M1 macrophages".
3. The procedure of evaluation of the transcriptome sequencing study (Figure 5A), biological process (Figure 5B) and KEGG pathway analysis should be described in the methods section at least briefly.
4. CD-HFD should be explained in the abbreviations sections.
5. Bodyweight development should be shown for MCD- and CD-HFD-fed mice, for the last group especially before and after DNT transfer.
6. Especially graphs for mRNA analysis could be bigger for a better legibility.
7. Picture size of histological slides should be enlarged for better perceptibility.

8. Regarding supplemental Figure 1F: Authors show Glucose levels after 12 weeks. Does this show the time point before DNT transfer? Please clarify in the description.
9. Regarding supplemental Figure 1: Please state the number of samples "n" for CD-HFD-fed illustrations.
10. In several graphs the lines showing significant differences are not of the same strength. Authors should standardize their illustrations.
11. On page 15 (discussion) the word "of" is missing in the first sentence.
12. IFN- γ is spelled as "IFN-r", e.g. in Figure 2 E and supplemental Figure 1D. Please change this into correct spelling.
13. In Figure 4C macrophages were co-cultured with DNT and TNF α secretion was examined. Please describe more precisely, if this is only examined from M1 macrophages. Moreover it would also be interesting to test for IL-6 and IFN- γ secretion.

Reviewer #1 (Remarks to the Author):

The findings of this paper are rather surprising given that DN T cells are generally considered a pro-inflammatory subset within the T cell compartment both in men and mice (1-4). DN T cells exhibit greatly elevated metabolic activity, which is characterized by accumulation of oxidative stress generating mitochondria and activation of the mechanistic target of rapamycin (5). The expansion of DN T cells has been widely linked to pathogenesis of lupus both in mice and humans (5-11). Importantly, lupus patients and mice also exhibit oxidative stress-driven inflammation of the liver which benefits from depletion of DN T cells by using antioxidants, such as acetylcysteine or rapamycin (12,13). DN T cells signal through the TCR and PD1 which have not been evaluated or even pictured in the proposed mechanism of action in Figure 7. Likewise, mTOR activation is critical for DN T cell function and T-cell lineage specification in general, which is also missing from the authors' concept.

Importantly, rapamycin also blocks diet-induced NALFD (Hepatology. 2014 Nov;60(5):1581-92). Therefore, it would be important to evaluate TCR signaling and use rapamycin as a positive control for the clinical efficacy of DN T cell therapy in steatohepatitis.

1. DN T cells are generally considered a pro-inflammatory subset within the T cell compartment both in men and mice

Answer: We thank the reviewer for raising these important questions.

Although DN T cells (DNT) only account for 1%-3% of total T lymphocytes in the peripheral blood and lymph organs of humans and mice, an increasing number of studies have proven that this rare T cell population plays critical and diverse roles in the immune system.

Some studies suggest that DNT (CD3⁺TCR $\alpha\beta$ ⁺CD4⁻CD8⁻NK1.1⁻(mouse)/CD56⁻(human)) are essential for maintaining immune system homeostasis with antigen specificity[1, 2]. DNT highly express perforin, granzyme B and FasL and exhibit strong

suppressive activity toward CD4⁺ and CD8⁺ T cells [3-6], as well as B cells[7, 8], dendritic cells[9], and NK cells[10], which are capable of suppressing the immune response and providing significant protection against allograft rejection, graft-versus-host disease (GVHD), and autoimmune diseases[3, 5, 11-14].

Indeed, as the reviewer mentioned, some reports do suggest that DNT are involved in systemic inflammation and tissue damage under autoimmune or inflammatory conditions. In patients with systemic lupus erythematosus (SLE)[15-17] or primary Sjögren's syndrome (pSS)[18, 19], DNT that expands in peripheral blood and inflamed tissues are the major producers of the proinflammatory cytokine IL-17 and are believed to be pathogenic in these autoimmune diseases.

Overall, current studies strongly support the notion that DNT comprise a rare but heterogeneous T lymphocyte subset.

1) To better understand the heterogeneity of DNT, we have already performed single-cell RNA sequencing of naïve mouse DNT (This work is under review at iScience now). As shown in Figure 1, naïve DNT include 2 subsets. One naïve DNT cluster highly expressed *Il17a*, suggesting these DNT are pro-inflammatory. However, single-cell RNA sequencing did show that the other DNT cluster which have no *Il17a* expression but highly expressed *Prf1*. This observation indicated that naïve DNT include at least two subsets, a pro-inflammatory subset and a non-inflammatory subset (might be regulatory DNT). These different DNT clusters may be activated and play proinflammatory or anti-inflammatory function.

Figure 1. UMAP plots showing proinflammation and anti-inflammation gene expression in naïve DNT clusters.

2) The DNT we used in this study are converted from CD4⁺ T cells and are defined as one type of regulatory T cells. These DNT do not express IL-2, IL-4 and IL-17 [3, 20], are not like IL-17 producing CD8 derived pro-inflammatory DNT.

In order to better understand the roles of these CD4 derived DNT, we firstly detected the activated CD4⁺ T cells and DNT cytokines secretion and immunosuppressive molecule expression *in vitro*. As shown in Figure 2A-C, compared with activated CD4⁺ T cells, the converted DNT had extremely low or no secretion of IL-2, IL-4, IL-6, IL-10, IL-13, IL-17A, IL-21 and TGF- β , except for IFN- γ . However, compared with CD4⁺ T cells, the converted DNT highly expressed Granzyme B, the key molecular of regulatory DNT.

As the reviewer suggested, we also transferred converted DNT into MCD-fed mice. Four weeks later, we detected the ability of DNT to produce cytokines *in vivo*. As shown in Figure 2D, these DNT also had extremely low or no secretion of IL-2, IL-4, IL-6, IL-10 and IL-17, including IFN- γ *in vivo*, but highly expressed Granzyme B. These observations indicated that the DNT are regulatory immune cells and are not proinflammatory cells. These results were also shown in the revised manuscript (Figure S6A-D).

To better distinguish the DNT we used in this study with proinflammatory DNT, we have added a section in the methods to describe the detail of the generation of converted DNT.

“Conversion of DNT *in vitro*

The conversion of CD4⁺ T cells to DNT was described previously. Briefly, CD4⁺CD25⁻ T cells were isolated from CD45.1 positive C57BL/6 mice spleens and lymph nodes. Then, the purified CD4⁺CD25⁻ T cells were co-cultured with C57BL/6 mature DCs and rIL-2 (50 ng/ml) for 7 days. Converted DNT were sorted from culture through a fluorescence-activated cell sorter (FACS Aria II, BD Biosciences).”

We also change the introduction in our revised manuscript as below to make it clear.

“Double negative T cells (DNT), which are characterized as TCR $\alpha\beta$ ⁺CD3⁺CD4⁻CD8⁻NK1.1⁻ in mice and TCR $\alpha\beta$ ⁺CD3⁺CD4⁻CD8⁻CD56⁻ in humans, comprise only 1-5% of peripheral T lymphocytes in mice and humans[21, 22]. An increasing number of studies have proven that this rare T cell population plays critical and diverse roles in the immune system. Some reports showed that DNT were involved in systemic inflammation and tissue damage under autoimmune or inflammatory conditions[18, 23-25]. However, studies also suggest that DNT are essential for maintaining immune system homeostasis with antigen specificity[26, 27]. Young NOD mice that display a high proportion of splenic DNT are potentially resistant to diabetes[28]. Graft-versus-host disease (GVHD) does not develop in individuals with greater than 1% DNT[29]. We have identified a differentiation pathway from CD4⁺ T cells to DNT in vitro and in vivo[3, 20]. Ex vivo-generated CD4⁺ T cell-converted DNT could prevent or reverse the onset of type 1 diabetes, allergic asthma, and induce allograft tolerance[3, 30-32].”

Figure 2. The cytokines secretion and immunoregulatory molecules expression profile of CD4 derived DNT *in vitro* and *in vivo*.

2. mTOR activation is critical for DN T cell function and T-cell lineage specification in general, which is also missing from the authors' concept.

Answer: We thank the reviewer for pointing this out. Several studies reported that mTOR activation is critical for DNT function, especially for the IL-4 production and necrotic death in systemic lupus erythematosus[33]. In order to determine the roles of mTOR signal pathway in IL-10 stimulated DNT, we also examined pi-mTOR expression of DNT with or without IL-10 stimulation *in vitro*. As shown in Figure 3A, IL-10 stimulation did not increase pi-mTOR expression, which indicated mTOR signal pathway

might not important for DNT during IL-10 stimulation. Meanwhile, we also found IL-10 stimulation did not affect DNT cytokines secretion. But rapamycin, the inhibitor of mTOR signal pathway, could suppress IFN- γ secretion in DNT (Figure 3B).

3. DN T cells signal through the TCR and PD1 which have not been evaluated or even pictured in the proposed mechanism of action in Figure 7.

Answer: Several studies reported that, within DNT, the PD-1⁺ subset generates the majority of pro-inflammatory cytokines[34]. Therefore, we also detected PD-1 expression with or without IL-10 stimulation. As shown in Figure 3C-D, IL-10 stimulation could markedly increase PD-1 expression. Then we also used PD-L1/Fc to activate PD-L1/PD-1 signaling and found PD-L1/Fc did not affect the cytokine secretion of DNT (Figure 3E). These observations indicated PD-L1/PD-1 signaling might not play important roles in CD4 converted DNT cytokines production.

Figure 3. The cytokines secretion of converted DNT after mTOR or PD-1 signal activation.

References

1. Juvet, S.C. and L. Zhang, *Double negative regulatory T cells in transplantation and autoimmunity: recent progress and future directions*. Journal of Molecular Cell Biology, 2012. **4**(1): p. 48-58.
2. Hillhouse, E.E., J.S. Delisle, and S. Lesage, *Immunoregulatory CD4(-) CD8(-) T cells as a potential therapeutic tool for transplantation, autoimmunity, and cancer*. Frontiers in Immunology, 2013. **4**.
3. Zhang, D., et al., *New differentiation pathway for double-negative regulatory T cells that regulates the magnitude of immune responses*. Blood, 2007. **109**(9): p. 4071-4079.
4. Chen, W.H., et al., *Both infiltrating regulatory T cells and insufficient antigen presentation are involved in long-term cardiac xenograft survival*. Journal of Immunology, 2007. **179**(3): p. 1542-1548.
5. Zhang, D., et al., *Adoptive cell therapy using antigen-specific CD4(-)CD8(-) T regulatory cells to prevent autoimmune diabetes and promote islet allograft survival in NOD mice*. Diabetologia, 2011. **54**(8): p. 2082-2092.
6. Fischer, K., et al., *Isolation and characterization of human antigen-specific TCR alpha beta(+) CD4(-)CD8(-) double negative regulatory T cells*. Blood, 2005. **106**(11): p. 924a-924a.
7. Zhang, Z.X., et al., *Double-negative T cells, activated by xenoantigen, lyse autologous B and T cells using a perforin/granzyme-dependent, fas-fas ligand-independent pathway*. Journal of Immunology, 2006. **177**(10): p. 6920-6929.
8. Li, W.X., et al., *Ex vivo converted double negative T cells suppress activated B cells*. International Immunopharmacology, 2014. **20**(1): p. 164-169.
9. Gao, J.F., et al., *Regulation of antigen-expressing dendritic cells by double negative regulatory T cells*. European Journal of Immunology, 2011. **41**(9): p. 2699-2708.
10. He, K.M., et al., *Donor double-negative Treg promote allogeneic mixed chimerism and tolerance*. European Journal of Immunology, 2007. **37**(12): p. 3455-3466.
11. Zhang, Z.X., et al., *Identification of a previously unknown antigen-specific regulatory T cell and its mechanism of suppression*. Nature Medicine, 2000. **6**(7): p. 782-789.
12. Young, K.J., et al., *Inhibition of graft-versus-host disease by double-negative regulatory T cells*. Journal of Immunology, 2003. **171**(1): p. 134-141.
13. McIver, Z., et al., *Double-negative regulatory T cells induce allotolerance when expanded after allogeneic haematopoietic stem cell transplantation*. British Journal of Haematology, 2008. **141**(2): p. 170-178.
14. Duncan, B., et al., *Double Negative (CD3(+)/4(-)8(-)) TCR alpha beta Splenic Cells from Young NOD Mice Provide Long-Lasting Protection against Type 1 Diabetes*. Plos One, 2010. **5**(7).
15. Crispin, J.C., et al., *Expanded Double Negative T Cells in Patients with Systemic Lupus Erythematosus Produce IL-17 and Infiltrate the Kidneys*. Journal of Immunology, 2008. **181**(12): p. 8761-8766.
16. Crispin, J.C. and G.C. Tsokos, *Human TCR-alpha beta(+) CD4(-) CD8(-) T Cells Can Derive from CD8(+) T Cells and Display an Inflammatory Effector Phenotype*. Journal of Immunology, 2009. **183**(7): p. 4675-4681.
17. Doreau, A., et al., *Interleukin 17 acts in synergy with B cell-activating factor to influence B cell*

- biology and the pathophysiology of systemic lupus erythematosus*. Nature Immunology, 2009. **10**(7): p. 778-U142.
18. Alunno, A., et al., *CD4(-)CD8(-) T-cells in primary Sjogren's syndrome: Association with the extent of glandular involvement*. Journal of Autoimmunity, 2014. **51**: p. 38-43.
 19. Alunno, A., et al., *IL-17-producing CD4(-)CD8(-) T cells are expanded in the peripheral blood, infiltrate salivary glands and are resistant to corticosteroids in patients with primary Sjogren's syndrome*. Annals of the Rheumatic Diseases, 2013. **72**(2): p. 286-292.
 20. Zhao, X., et al., *A novel differentiation pathway from CD4(+) T cells to CD4(-) T cells for maintaining immune system homeostasis*. Cell Death & Disease, 2016. **7**.
 21. Zhang, Z.X., K. Young, and L. Zhang, *CD3(+)CD4(-)CD8(-) alpha beta-TCR+ T cell as immune regulatory cell*. Journal of Molecular Medicine-Jmm, 2001. **79**(8): p. 419-427.
 22. Fischer, K., et al., *Isolation and characterization of human antigen-specific TCR alpha beta(+) CD4(-)CD8(-) double-negative regulatory T cells*. Blood, 2005. **105**(7): p. 2828-2835.
 23. Crispin, J.C., et al., *Expanded double negative T cells in patients with systemic lupus erythematosus produce IL-17 and infiltrate the kidneys*. J Immunol, 2008. **181**(12): p. 8761-6.
 24. Crispin, J.C. and G.C. Tsokos, *Human TCR-alpha beta+ CD4- CD8- T cells can derive from CD8+ T cells and display an inflammatory effector phenotype*. J Immunol, 2009. **183**(7): p. 4675-81.
 25. Doreau, A., et al., *Interleukin 17 acts in synergy with B cell-activating factor to influence B cell biology and the pathophysiology of systemic lupus erythematosus*. Nat Immunol, 2009. **10**(7): p. 778-85.
 26. Hillhouse, E.E., J.S. Delisle, and S. Lesage, *Immunoregulatory CD4(-)CD8(-) T cells as a potential therapeutic tool for transplantation, autoimmunity, and cancer*. Front Immunol, 2013. **4**: p. 6.
 27. Juvet, S.C. and L. Zhang, *Double negative regulatory T cells in transplantation and autoimmunity: recent progress and future directions*. J Mol Cell Biol, 2012. **4**(1): p. 48-58.
 28. Duncan, B., et al., *Double negative (CD3+ 4- 8-) TCR alphabeta splenic cells from young NOD mice provide long-lasting protection against type 1 diabetes*. PLoS One, 2010. **5**(7): p. e11427.
 29. McIver, Z., et al., *Double-negative regulatory T cells induce allotolerance when expanded after allogeneic haematopoietic stem cell transplantation*. Br J Haematol, 2008. **141**(2): p. 170-8.
 30. Zhang, D., et al., *Adoptive cell therapy using antigen-specific CD4⁻CD8⁻T regulatory cells to prevent autoimmune diabetes and promote islet allograft survival in NOD mice*. Diabetologia, 2011. **54**(8): p. 2082-92.
 31. Liu, T., et al., *Combination of double negative T cells and anti-thymocyte serum reverses type 1 diabetes in NOD mice*. J Transl Med, 2016. **14**: p. 57.
 32. Tian, D., et al., *Double negative T cells mediate Lag3-dependent antigen-specific protection in allergic asthma*. Nature Communications, 2019. **10**.
 33. Lai, Z.W., et al., *Mechanistic Target of Rapamycin Activation Triggers IL-4 Production and Necrotic Death of Double-Negative T Cells in Patients with Systemic Lupus Erythematosus*. Journal of Immunology, 2013. **191**(5): p. 2236-2246.
 34. Rodriguez-Rodriguez, N., et al., *Pro-inflammatory self-reactive T cells are found within murine TCR-alpha beta(+)CD4(-)CD8(-)PD-1(+) cells*. European Journal of Immunology, 2016. **46**(6): p. 1383-1391.

4. Importantly, rapamycin also blocks diet-induced NALFD (Hepatology. 2014 Nov;60(5):1581-92). Therefore, it would be important to evaluate TCR signaling and use rapamycin as a positive control for the clinical efficacy of DN T cell therapy in steatohepatitis.

Answer: We thank the reviewer for the insightful comments. As the reviewer suggested, we also served rapamycin as a positive control in this study. In the MCD-fed NASH model, we transferred 5×10^6 CD45.1-positive DNT (purity > 97%) into the C57BL/6 mice, and then fed the MCD for 4 weeks. Meanwhile, we also treated the MCD-fed mice with rapamycin (0.5mg/kg/day, intraperitoneally) as a positive control for 4 weeks.

As shown in Figure 4, rapamycin could restrict fat accumulation, lobular inflammation, and focal necrosis (Figure 4A and 4B in this response letter). Compared with control mice, MCD-fed mice with rapamycin treatment showed significantly decreased plasma ALT, AST and TG levels (Figure 4C in this response letter). These observations were similar with that in MCD-fed mice with DNT transferred. These results were also shown in the revised manuscript (Figure S2A-2C).

Meanwhile, rapamycin treatment could also suppress liver-infiltrating monocyte-derived macrophages. While rapamycin could not only downregulate M1 and Ly6C^{hi} macrophages proportions, but also inhibit M2 macrophages proportions (Figure 4D and 4E in this response letter).

The rapamycin remarkably decreased macrophage TNF- α secretion, and CD4⁺ T cells IL-17 production in MCD-fed mice, which was similar with DNT treatment. However, rapamycin treatment could also significantly inhibit T cells IFN- γ , IL-13 secretion, and macrophages IL-6, IL-10 production, which were not observed in DNT transferred mice (Figure 4F-I in this response letter).

Figure 4. Compared with DNT transferred, rapamycin treatment could also delay NASH development in MCD-fed mice.

Specific comments:

Figure 1. This figure supports the contention that a single transfer of DN T cells prevented diet-induced obesity, insulin resistance and NASH development. Apparently, mice that received DNT therapy had lower plasma IFN- γ , TNF- α , IL-6, IL-9, IL-17A and IL-17F levels (Figure 1K). Fibrosis-related genes, such as SMA, Col1a1 and Col3a1, were also downregulated in the livers of HFD-fed mice that underwent DNT transfer (Figure 1L). This figure should evaluate the functionality of DN T cells and their ability to produce pro-inflammatory cytokines.

Answer: We thank the reviewer for the insightful comments. As the reviewer suggested, we have detected the cytokine secretion and immune regulatory molecules expression of DNT *in vivo* and *in vitro*.

As shown in Figure 2A-C in response letter, compared with activated CD4⁺ T cells, the converted DNT had extremely low or no secretion of IL-2, IL-4, IL-6, IL-10, IL-13, IL-17, IL-21 and TGF- β , except for IFN- γ . However, compared with CD4⁺ T cells, the converted DNT highly expressed Granzyme B, the key molecular of regulatory DNT. Furthermore, we also detected the ability of DNT to produce cytokines *in vivo*. As shown in Figure 2D, these DNT also had extremely low or no secretion of IL-2, IL-4, IL-6, IL-10 and IL-17, including IFN- γ *in vivo*, but highly expressed Granzyme B. These observations indicated that these converted DNT are regulatory immune cells and are not proinflammatory cells. These results were also shown in the revised manuscript (Figure S6A-D).

Figure 2. T cell subsets, including CD4, CD8, DN, and Treg cells and NK cell subsets and their intracellular cytokine production profiles should be presented as flow cytometry dot plots including the gating strategy. It would be important to systematically present cell populations from liver, spleen, and lymph node of the same

mice.

Answer: We appreciate the reviewer's suggestion.

1) As the reviewer suggested, we presented the intracellular cytokine production profiles in different cell subsets as flow cytometry dot plots, and added the gating strategy in Figure S4 in revised manuscript. Meanwhile, we also detected the IFN- γ secretion in different cell subsets, such as CD4⁺, CD8⁺, DNT, NK and NKT. As shown in Figure 5 in this response letter, the production of IFN- γ in CD4⁺, CD8⁺, NK cells and IL-17 in CD4⁺ cells were more abundant in HFD-fed mice, while the mice that received DNT exhibited a marked decrease in the percentage of IL-17-producing CD4 T cells. These results were also shown in the revised manuscript (Figure 2E-F).

Figure 5. The cytokines secretion in T cells, NK cells and NKT cells.

2) Then we also presented cell population from liver, adipose, spleen, blood and draining lymph node. As shown in Figure 6 in this response letter, the proportions of CD4⁺ and CD8⁺ T cells markedly increased in liver tissues, adipose tissues, spleen and draining lymph nodes of HFD-fed mice. The transferring of DNT significantly reduced the proportion of CD4⁺ T cells, but not CD8⁺, NK and NKT cells, in liver tissues, spleen, and draining lymph nodes. Similar results were also found in adipose tissue, DNT transfer significantly decreased CD4⁺ and CD8⁺ T cell proportions. Meanwhile, we also found HFD-fed mice had an increased

proportion of macrophages in liver tissues, adipose tissues and draining lymph nodes, and DNT markedly inhibited macrophage liver and adipose tissue infiltration. However, there were no significant differences in the proportions of neutrophils in liver tissues, adipose tissues, spleen, blood and lymph nodes with or without DNT transfer (Figure 2A, 2B, 2H, Figure 3A, 3I and Figure S7A-7C in revised manuscript).

Figure 6. The proportions of lymphocytes in different tissues of the same mice.

Abstract: The statement “there are currently no approved treatments for NASH/NAFLD” is too vague and it should be removed from the abstract. One might consider replacement with effective treatment. However, this statement rather belongs to the discussion clarifying what “approved” means. Again, it would be important to use rapamycin as a positive control which eliminates proinflammatory T cell development.

Answer: We thank the reviewer for his suggestion, and we have removed this sentence from the abstract. Then, we also served rapamycin as a positive control in my study, and these results were showed in Figure 4 in this response letter.

Reviewer #2 (Remarks to the Author):

The manuscript by Sun et al. addresses the potential of adoptively transferred double-negative T cells to prevent the development of NASH in a mouse model based on feeding a high-fat diet. Moreover, they describe a potential mechanism regulating the suppressive activity of these cells, which seems to be based on interaction between the inhibitory receptor NKG2A and Qa-1.

The study is of potential interest notably in the mechanistic part; however, it is rather weak in its in vivo part.

Specific comments:

1. The NASH model is problematic. Although I can appreciate the effect of the transferred DN T cells on steatosis and metabolic parameters, it is hard to see the effect on inflammation. Mainly, because the degree of inflammation and liver damage in the high-fat diet group not receiving DN T cells is very mild. The ALT/AST levels are only slightly increased. The HE staining does not show significant infiltration. In other words: there is almost no hepatitis. Therefore, I find it difficult to follow the authors' claim that DN T cells could be used for NASH treatment.

(By the way, the authors should take care that the HE sections that are compared between the groups are similar with respect to size and distribution of portal fields and central veins; now they compare sections showing only veins with sections showing mostly portal tracts.)

Answer: We thank the reviewer for his suggestion.

The reviewer thought that the degree of inflammation and liver damage in the high-fat diet group is very mild. We do agree with the reviewer, that was why we used 3 different diet induced NASH models in our study.

There are several dietary models of nonalcoholic fatty liver disease, the choice of models is somehow difficult for NASH studies[35]. ① MCD diets produce the most severe phenotype in the shortest timeframe, have been broadly used over 40 years, and could induce hepatic steatosis in mice by 2-4 weeks and this progresses to inflammation and fibrosis shortly thereafter. However, unlike human or high-fat diet (HFD)-induced rodent models of NAFLD, rodents fed MCD diets lose weight and do not become insulin resistant. ② HFDs are well-known to increase body weight and induce insulin resistance in mice. However, HFDs can also induce mild liver steatosis and steatohepatitis, although at lower levels compared with MCD. ③ Another less applied model is CD-HFD induced NASH model. CD-HFD is reported to exacerbate liver steatosis and steatohepatitis without weight loss in mice, CD-HFD would also attenuates insulin resistance which is similar to MCD but is not seen in human NAFLD or HFD induced NAFLD[36]. Based on these reasons, we used above 3 different NASH models to evaluate the protection of DNT. In HFD, MCD and CD-HFD models, DNT clearly suppressed liver inflammation and damage.

As the reviewer suggested, we re-select field of histological slides and take pictures with 100× magnification to contain portal fields and central veins in each view. As shown in Figure 7, 8 and 9 in this response letter, we could see liver inflammation and liver damage in HFD, MCD and CD-HFD-fed mice.

Meanwhile, we also blindly evaluated the severity of NASH by 3 experienced pathologists using the NAS score. In HFD, MCD and CD-HFD-fed mice, transferred with DNT significantly decreased NAS score (HFD mice, 5.67 ± 0.2108 vs 2.00 ± 0.2582 , $p < 0.0001$; MCD mice, 6.67 ± 0.2117 vs 4.00 ± 0.3651 , $p < 0.0001$; CD-HFD mice, 6.75 ± 0.2500 vs 4.25 ± 0.2500 , $p < 0.0001$). Most studies have used threshold values of the NAS, specifically $NAS \geq 5$, as a surrogate for the histologic diagnosis of NASH. Therefore, we believe the induction of NASH in this study was successful. Meanwhile, we also quantified the positive area of Oli Red O staining, and observed

the transferred of DNT could decrease liver fat accumulation in NASH model. These results were also shown in the revised manuscript (Figure 1J-1K, Figure S1C and S3C).

Figure 7. Liver histology in HFD-fed mice with or without DNT transfer.

Figure 8. Liver histology in MCD-fed mice with or without DNT transfer.

Figure 9. Liver histology in CD-HFD-fed mice with or without DNT transfer.

2. Along the same line: if the authors wish to claim an treatment effect on fibrosis, they should show it not merely by PCR for some markers, but by more adequate assay, such as Sirius red staining, hydroxyproline content, aSMA immunostaining or the like.

Answer: We thank the reviewer for the insightful comments.

As the reviewer suggested, we determined the DNT on fibrosis through Sirius red staining, α -SMA staining and hydroxyproline content detection. As shown in Figure 7-9 in this response letter, DNT transfer significantly decreased the staining of Sirius red and α -SMA in HFD, MCD and CD-HFD-fed mice. Meanwhile, we also determined α -SMA expression in protein levels through western blotting, and similar results were found after DNT transfer in MCD-fed mice (Figure 10A-B in this response letter).

These results were also shown in the revised manuscript (Figure 1J, 1K, Figure S1C, S1D and Figure S3C).

Hydroxyproline is a major component of fibrillar collagen. As the reviewer's suggestion, we also determined hydroxyproline content in NASH liver tissues. As shown in Figure 10C, hepatic hydroxyproline content increased in HFD, MCD and CD-HFD-fed mice, and DNT transferred downregulated the levels in these mice. These results were also shown in the revised manuscript (Figure 1L, Figure S1F and Figure S3E).

Figure 10. The examination of α -SMA and hydroxyproline levels in liver tissues with or without DNT transfer.

3. The time-point of CD45.1 analysis after transfer of DN T cells (Fig 1C) is not clear. Actually, one would like to see several time-points to get an idea of the dynamics of survival and tissue distribution.

Answer: We thank the reviewer for the insightful comments. In order to confirm the transferred DNT dynamics of survival and tissue distribution *in vivo*, we transferred CD45.1⁺ DNT into MCD-fed mice for 4 weeks. As shown in Figure 11 in this response letter, we detected CD45.1⁺ DNT in blood, spleen, liver tissues and different lymph nodes every week. We found CD45.1⁺ DNT mainly accumulated in liver tissues and draining lymph nodes. These results were also shown in the revised manuscript (Figure S5A-B).

As the reviewer suggested, we also detected the apoptosis and proliferation with Annexin V and Ki67 staining of transferred DNT in liver tissues. We mainly detected the DNT survival at 3 and 4 weeks, because the proportions of CD45.1⁺ DNT in the liver at 1 and 2 weeks after adoptive transfer were too low to analysis. As shown in Figure 12A and 12B in this response letter, the percentage of Annexin V⁺ DNT were less than 15%, and proportion of Ki67⁺ DNT were more than 40% at 3 weeks and 4 weeks after DNT adoptive transfer, which indicated that the transferred DNT were with low apoptosis and high proliferation in liver tissues. Meanwhile, these DNT has also maintained a CD4⁺CD8⁻, suggesting these cells were stable *in vivo* (Figure 12C). These results were also shown in the revised manuscript (Figure S6E-G).

Figure 11. The tissue distribution of transferred CD45.1⁺ DNT in MCD-fed mice.

Figure 12. The survival and CD4, CD8 expression of transferred DNT *in vivo*.

4. The method of DN T cell generation is not described. Mere reference to a previous paper is insufficient (Ref.16), as there are several alternative methods described. The details of the applied cytokines, APCs, number and chronology of re-stimulations, and the like are essential, but lacking.

Answer: We thank the reviewer for the kind suggestion. As the reviewer suggested, we have added a section in the methods to describe the detail of generating of DNT.

“Conversion of DNT *in vitro*

The conversion of CD4⁺ T cells to DNT was described previously. Briefly, CD4⁺CD25⁻ T cells were isolated from CD45.1 positive C57BL/6 mice spleens and lymph nodes. Then, the purified CD4⁺CD25⁻ T cells were co-cultured with C57BL/6 mature DCs and rmIL-2 (50 ng/ml) for 7 days. Converted DNT were sorted from culture through a fluorescence-activated cell sorter (FACS Aria II, BD Biosciences).”

5. On page 9 in the context of Figure 2, the authors conclude ‘that a single transfer of DNT to HFD-fed mice could reduce hepatic inflammation and NASH development by decreasing hepatic CD4⁺ T-cell proportion and survival and Th17 cell differentiation’. On the basis of the current data, I think that this is merely an association. To really sustain

their statement, it would require some kind of inhibition experiment, which is however not provided.

Answer: We thank the reviewer for the insightful comments. We cocultured DNT and Th17 cells and performed an inhibition experiment *in vitro*.

Naïve CD45.1 positive CD4⁺ T cells (3×10^5 /well) were isolated from mice spleens, and stimulated with anti-CD3 (2 μ g/ml), anti-CD28 (1 μ g/ml), IL-6 (2 ng/ml), TGF- β (1 ng/ml), anti-IL-4 (2 μ g/ml) and anti-IFN- γ (5 μ g/ml) in 96-well flat-bottom culture plates for 3 days for Th17 cells differentiation. After 3 days culture, 0.75×10^5 /well and 3×10^5 /well converted CD45.2⁺ DNT were added into the culture for 24 hours. Then Annexin V staining and IL-17 intracellular staining were performed to detect the apoptosis and Th17 differentiation of CD45.1⁺ CD4⁺ cells. As shown in Figure 13, the converted DNT not only increased the apoptosis of CD4⁺ T cells, including Th17 cells, but also suppressed the IL-17 secretion of CD4⁺ T cells. These results were also shown in the revised manuscript (Figure S7F-G).

Figure 13. The converted DNT suppressed Th17 cells survival and differentiation.

6. The *in vitro* experiments showing the possible function of IL-10, NKG2A, and Qa-1 are interesting. However, to really claim a clear functional relevance, it will be necessary to perform inhibition experiments *in vivo*, e.g. using inhibitory antibodies to IL-10 or IL10R, or Qa-1. However, I am afraid that such experiments will again be limited by the inherent problems of the NASH model used by the authors.

Answer: We thank the reviewer for raising this important question. As the reviewer suggested, we performed the assay in MCD induced NASH model *in vivo*. For our treatment model, we treated the MCD-fed mice by intraperitoneal injection with 0.3 mg of anti-IL-10 Ab (JES5-2A5; BioXCell) every other day after DNT transfer for 4 weeks. And we detected the liver tissues inflammation and lymphocytes proportion at 4 weeks.

As shown in Figure 14A-C in this response letter, a single transfer of DNT could inhibit NASH development in MCD-fed mice. As expected, mice treated with IL-10 neutralizing antibody together with DNT showed increased plasma ALT levels, liver fat accumulation, inflammation and NASH development. Meanwhile, we also detected the proportion of M1, M2 macrophages after anti-IL-10 neutralizing antibody treatment. Compared with DNT treatment alone, combination DNT with anti-IL-10 neutralizing antibody could downregulate liver macrophages proportions, especially M2 macrophages after 4 weeks MCD feeding (Figure 14D-F). Then we also found transferred DNT with anti-IL-10 neutralizing antibody could increase M2 macrophages apoptosis and downregulate these cells proliferation at 4 weeks (Figure 14G-H). IL-10 neutralizing antibody injection could inhibit Bcl-2 and Bcl-xl expression in M2 macrophages at 4 weeks (Figure 14I-J). Furthermore, we also observed that IL-10 neutralizing antibody could downregulate NKG2A but not NKG2D expression in DNT at 4 weeks (Figure 14K). These data demonstrated that anti-IL-10 neutralizing antibody increased DNT cytotoxicity toward M2 macrophages through inhibiting DNT NKG2A expression *in vivo*. These results were also shown in the revised manuscript (Figure 7).

Figure 14. Anti-IL-10 neutralizing antibody administration increased DNT cytotoxicity to M2 macrophages *in vivo*.

7. Minor points are related to suboptimal gating strategy for DCs as MHC II/CD11c high (what about pDCs?), and M1/M2 based merely on CD206 and MHC II.

Answer: We thank the reviewer for this suggestion. In this study, we defined hepatic M1-like macrophages as $CD45^+Ly6G^-CD11b^{high}F4/80^{int}CD11C^+CD206^-$ and M2-like macrophages by $CD45^+Ly6G^-CD11b^{high}F4/80^{int}CD11C^-CD206^+$ respectively as previous reports [37-39], and found transferred DNT could downregulate M1 cells proportion, but not M2 cells (Figure 3B-3D in revised manuscript). As the reviewer suggested, we also defined hepatic M1 macrophage and M2 macrophages by $CD45^+Ly6G^-CD11b^{high}F4/80^{int}MHC\ II^+CD206^-$ and $CD45^+Ly6G^-CD11b^{high}F4/80^{int}MHC\ II^-CD206^+$ respectively, similar results were also found in Figure 15A-B in this response letter. These results were also shown in the revised manuscript (Figure 3B and 3E).

Meanwhile, we also defined pDCs by $CD45^+CD3^-CD19^+B220^+BST-2^+CD11C^{int}$ as previous studies [40-42] and found transferred DNT had no effects on these cell (Figure 15C-D in this response letter). These results were also shown in the revised manuscript (Figure S9A).

Figure 15. The proportions of M1/M2 macrophages and pDCs in NASH models with or without DNT transferred.

8. I have some doubt that the use of the t test is really appropriate, as I do not think that it normally distributed.

Answer: We thank the reviewer for the insightful comments. As the reviewer suggested, we have provided statistical analysis which performed in our study in detail. Values are expressed as the mean \pm standard deviation (SD). In this study, the normal distribution of variables was tested with the Shapiro-Wilk test. Differences between groups were compared by t test for normal variables and Mann-Whitney test for nonnormal variables. P values <0.05 were considered significant.

References

35. Larter, C.Z. and M.M. Yeh, Animal models of NASH: Getting both pathology and metabolic context right. *Journal of Gastroenterology and Hepatology*, 2008. **23**(11): p. 1635-1648.
36. Raubenheimer, P.J., M.J. Nyirenda, and B.R. Walker, A choline-deficient diet exacerbates fatty liver but attenuates insulin resistance and glucose intolerance in mice fed a high-fat diet. *Diabetes*, 2006. **55**(7): p. 2015-2020.
37. Fen, Z.G., et al., DPP-4 Inhibition by Linagliptin Attenuates Obesity-Related Inflammation and Insulin Resistance by Regulating M1/M2 Macrophage Polarization. *Diabetes*, 2016. **65**(10): p. 2966-2979.
38. Ono, Y., et al., CD11c+M1-like macrophages (M Phi s) but not CD206+M2-like M Phi are involved in folliculogenesis in mice ovary. *Scientific Reports*, 2018. **8**.
39. Cabalen, M.E., et al., Chronic Trypanosoma cruzi infection potentiates adipose tissue macrophage polarization toward an anti-inflammatory M2 phenotype and contributes to diabetes progression in a diet-induced obesity model. *Oncotarget*, 2016. **7**(12): p. 13400-13415.
40. Arimura, K., et al., Crucial role of plasmacytoid dendritic cells in the development of acute colitis through the regulation of intestinal inflammation. *Mucosal Immunology*, 2017. **10**(4): p. 957-970.
41. Hansen, L., et al., E2-2 Dependent Plasmacytoid Dendritic Cells Control Autoimmune Diabetes. *Plos One*, 2015. **10**(12).
42. Liao, X.F., et al., Cutting Edge: Plasmacytoid Dendritic Cells in Late-Stage Lupus Mice Defective in Producing IFN-alpha. *Journal of Immunology*, 2015. **195**(10): p. 4578-4582.

Reviewer #3 (Remarks to the Author):

In their present study, the authors aimed to find a potential new therapy for NASH by allograft transfer of double negative T-cells (DNT) in steatohepatitis mouse models. By transfer of DNT into mice with NASH generated by different dietary models they found that DNT ameliorates inflammation and consequently improved NASH and glucose tolerance.

The main target of DNT was a reduction in inflammatory M1 macrophages. Subsequent cell culture analysis suggest that M2 macrophages are protective by IL-10 secretion via NKG2A/Qa1b signaling.

To address their query the authors used a well-structured approach and study design. Background information on their main methods was presented and results and evidence were conclusively constituted. Different NASH-models and cell culture stimulation approaches as well as cohesive methods like mRNA levels and protein expression were used to underline their findings.

Although the results are convincing, some major and minor points should be addressed to underline the study outcome.

Major points:

1. Liver histology should be further investigated and quantified. Specifically severity of NASH has to be evaluated by a blinded and experienced pathologist, e.g. using the international approved NAS score. Moreover, fibrosis must be measured by Sirius Red staining and lipomatosis should also be addressed in the HFD-fed mice by Oil Red O staining. All staining should be quantified. Differences in fibrosis related protein expression should be also affirmed by western blot analysis.

Answer: We thank the reviewer for the insightful suggestion.

As the reviewer's suggestion, we blindly evaluated the severity of NASH by 3 experienced pathologists using NAS score. As shown in Figure 7, 8 and 9, in HFD, MCD and CD-HFD-fed mice, transferring with DNT significantly decreased NAS

score (HFD mice, 5.67 ± 0.2108 vs 2.00 ± 0.2582 , $p < 0.0001$; MCD mice, 6.67 ± 0.2117 vs 4.00 ± 0.3651 , $p < 0.0001$; CD-HFD mice, 6.75 ± 0.2500 vs 4.25 ± 0.2500 , $p < 0.0001$). Meanwhile, we also quantified the positive area of Oli Red O staining, and observed the transferred of DNT could decrease liver fat accumulation in NASH model.

Furthermore, we also determined the DNT treatment on fibrosis through Sirius red staining, α -SMA staining and hydroxyproline content detection. As shown in Figure 7-9, DNT significantly decreased the positive area of Sirius red and α -SMA staining in HFD, MCD and CD-HFD-fed mice. We also determined α -SMA expression in protein levels through western blotting, and similar results were found after DNT transfer in MCD-fed mice (Figure 10A-10B in this response letter).

Hydroxyproline is a major component of fibrillar collagen. As the reviewer suggested, we determined hydroxyproline content in NASH livers tissues. As shown in Figure 10C, hepatic hydroxyproline content increased in HFD, MCD and CD-HFD-fed mice, and DNT downregulated the levels in these mice. These results were also shown in the revised manuscript (Figure 1J-1M, Figure S1C-1F and Figure S3C-3E).

Figure 7. Liver histology in HFD-fed mice with or without DNT transfer.

Figure 8. Liver histology in MCD-fed mice with or without DNT transfer.

Figure 9. Liver histology in CD-HFD-fed mice with or without DNT transfer.

Figure 10. The examination of α -SMA and hydroxyproline levels in liver tissues with or without DNT transferred.

2. The immune phenotype must be confirmed in liver histology by different immune staining methods. First of all the number of infiltrating leukocytes should be validated and quantified by CD45 staining (IHC and/or IF). Subpopulations could be addressed by CD3, CD20, CD11b and F4/80 stainings. Moreover, single-cell RNA sequencing (scRNA-seq) should be included to identify and differentiate phenotypes of M1 and M2 macrophages since the identification just by FACS markers can be improved.

Answer: We thank the reviewer for the kind suggestion.

As the reviewer's suggestion, we stained CD45, CD3, CD20 and CD11b in liver tissues of HFD-fed mice, and stained F4/80 in HFD, MCD and CD-HFD-fed mice. As shown in Figure 16 in this response letter, HFD-fed mice had more CD45, CD3, CD11b and F4/80-positive cells, and the transferring of DNT could downregulate these cell number,

except CD20-positive cells. F4/80 staining in liver tissues of MCD and CD-HFD-fed mice were also showed similar results. F4/80 positive cells increased heavily in MCD and CD-HFD-fed mice, and DNT cells transferred downregulated F4/80 positive cells numbers significantly. These results were also shown in the revised manuscript (Figure S7D-7E and S8A).

Although the identification of M1 and M2 macrophages just by FACS markers can be improved, most of the current studies are still using FACS to identify M1 and M2 macrophages [37-39, 43]. However, we totally agree with the reviewer's opinion, the single-cell RNA sequencing (scRNA-seq) would be helpful to identify and differentiate phenotypes of M1 and M2 macrophages, we will perform scRNA-seq in the future.

Figure 16. The immunohistochemistry of different immune cell markers in liver tissues of NASH models.

3. The same histological immune quantification should be done for VAT. Furthermore, size of adipocytes has to be quantified to show significant differences.

Answer: We thank the reviewer's suggestion. We have quantified adipocytes diameter and the numbers of infiltration immune cells in adipose tissues. As shown in Figure 17, transferring of DNT could decrease adipocytes size and lymphocyte adipose infiltration. HFD-fed mice had more CD45, CD3, CD20, CD11b and F4/80-positive cells, and DNT decrease these cells. These results were also shown in the revised manuscript (Figure 1J, 1K, and Figure S8B).

Figure 17. Transferred DNT decreased adipose tissues inflammation.

4. The authors describe in Figure 1C that DNT accumulate in Liver and VAT. In supplemental Figure 1A VAT is not shown but here draining lymph nodes are stated as major accumulation points. Can the authors explain this difference between those models? Why are lymph nodes not shown for HFD-fed mice? How do the authors clarify that it is only draining lymph nodes of the liver? Can they show data for non-draining lymph nodes and explain how they identified draining lymph nodes? Do they

also have data of VAT in this model?

Answer: We thank the reviewer for raising this important question.

It's our negligence and we added the data of DNT proportions in draining lymph nodes in HFD-fed mice. As shown in Figure 18, compared to liver tissues and VAT, these transferred DNT were also mainly accumulated in draining LN of recipient mice. Many studies reported that the draining LNs, which near the portal vein, is close proximity to the liver, and play important roles in liver immunoregulatory[44-47]. These results were also shown in the revised manuscript (Figure 1C).

Figure 18. The tissue distribution of transferred CD45.1⁺ DNT in HFD-fed mice.

Meanwhile, we also transferred CD45.1⁺ DNT into MCD-fed mice for 4 weeks. As shown in Figure 11 in this response letter, we detected CD45.1⁺ DNT in blood, spleen, liver tissues, adipose tissues and different lymph nodes every week. We also found CD45.1⁺ DNT mainly accumulated in liver tissues and draining lymph nodes, but not in other non-draining lymph nodes.

No CD45.1⁺ DNT accumulated in adipose tissues was observed in MCD-fed mice, which was different with the DNT highly accumulation in adipose tissues of HFD-fed mice. That may be because adipose tissues and insulin resistance play important roles

during HFD induced NASH development, while the mice fed MCD diets lose weight and do not become insulin resistant. These results were also shown in the revised manuscript (Figure S5A-B).

① MCD diets produce the most severe phenotype in the shortest timeframe, have been broadly used over 40 years, and could induce hepatic steatosis in mice by 2-4 weeks and this progresses to inflammation and fibrosis shortly thereafter. However, unlike human or high-fat diet (HFD)-induced rodent models of NAFLD, rodents fed MCD diets lose weight and do not become insulin resistant. ② HFDs are well-known to increase body weight and induce insulin resistance in mice. However, HFDs can also induce mild liver steatosis and steatohepatitis, although at lower levels compared with MCD. Based on these reasons, we used above 2 different NASH models to evaluate the accumulation of DNT in NASH mice.

Figure 11. The tissue distribution of transferred CD45.1⁺ DNT in MCD-fed mice.

5. Protein levels are shown by FACS analysis. The most important proteins in this study (e.g. IL-10, TNF α) should also be measured by western blot analysis to verify different protein levels.

Answer: We thank the reviewer for this kind suggestion. We have measured TNF- α , IL-10 expression in liver tissues by western blot. As shown in Figure 19 in this response letter, TNF- α expression increased in liver tissues of MCD-fed mice, and DNT downregulated TNF- α protein level. IL-10 expression decreased in MCD-fed mice, while DNT did not influence IL-10 protein level. These results were also shown in the revised manuscript (Figure S1D).

Figure 19. The protein levels of TNF- α and IL-10 in MCD-fed mice with or without DNT transfer.

6. In their abstract and discussion, authors state that among lymphocytes specifically TH17 cells are suppressed by DNT and confirm this in the animal models by FACS and mRNA levels. Can they explain or discuss the cause of this TH17-specific effect? Can they also contribute cell culture models to ensure this effect on TH17 cells?

Answer: We thank the reviewer for the insightful comments.

We cocultured DNT and Th17 cells and performed an inhibition experiment *in vitro*. Naïve CD45.1 positive CD4⁺ T cells (3×10^5 /well) were isolated from mice spleens, and stimulated with anti-CD3 (2 µg/ml), anti-CD28 (1 µg/ml), IL-6 (2 ng/ml), TGF-β (1 ng/ml), anti-IL-4 (2 µg/ml) and IFN-γ (5 µg/ml) in 96-well flat-bottom culture plates for 3 days for Th17 cells differentiation. After 3 days culture, 0.75×10^5 /well and 3×10^5 /well converted CD45.2⁺ DNT were added into the CD4⁺ T cells for 24 hours. Then Annexin V staining and IL-17 intracellular staining were performed to detect the apoptosis and Th17 differentiation of CD45.1⁺ CD4⁺ cells. As shown in Figure 13, the converted DNT could not only increase the apoptosis of CD4⁺ T cells, including Th17 cells, but also suppress the IL-17 secretion of CD4⁺ T cells. These results were also shown in the revised manuscript (Figure S7F-7G).

Figure 13. The converted DNT suppressed Th17 cells survival and differentiation.

7. The observed effect in vitro should be confirmed by in vivo experiments as well. Are macrophages and DNT in vivo affected when animals are treated with anti-IL-10 neutralizing mAb?

Answer: We thank the reviewer for raising this important question. As the reviewer suggested, we performed the assay in MCD induced NASH model *in vivo*. For our treatment model, we treated the MCD-fed mice by intraperitoneal injection with 0.3 mg of anti-IL-10 Ab (JES5-2A5; BioXCell) every other day after DNT transfer for 4 weeks. And we detected the liver tissues inflammation and lymphocytes proportion at 4 weeks.

As shown in Figure 14A-C in this response letter, a single transfer of DNT could inhibit NASH development in MCD-fed mice. As expected, mice treated with IL-10 neutralizing antibody together with DNT showed increased plasma ALT levels, liver fat accumulation, inflammation and NASH development. Meanwhile, we also detected

the proportion of M1, M2-like macrophages after anti-IL-10 neutralizing antibody treatment. Compared with DNT treatment alone, combination DNT with anti-IL-10 neutralizing antibody could downregulate liver macrophages proportions, especially M2-like macrophages after 4 weeks MCD feeding (Figure 14D-F). Then we also found transferred DNT with anti-IL-10 neutralizing antibody could increase M2 macrophages apoptosis and downregulate these cells proliferation at 4 weeks (Figure 14G-H). IL-10 neutralizing antibody injection could inhibit Bcl-2 and Bcl-xl expression in M2 macrophages at 4 weeks (Figure 14I-J). Furthermore, we also observed that IL-10 neutralizing antibody could downregulate NKG2A but not NKG2D expression in DNT at 4 weeks (Figure 14K). These data demonstrated that anti-IL-10 neutralizing antibody increased DNT cytotoxicity toward M2 macrophages through inhibiting DNT NKG2A expression *in vivo*. These results were also shown in the revised manuscript (Figure 7).

Figure 14. Treated with anti-IL-10 neutralizing antibody increased DNT cytotoxicity to M2 macrophages *in vivo*.

Minor points:

1. Authors discuss that the accumulation of DNT in liver and VAT could be due to CXCR3-CXCL10 interaction. Authors could measure CXCL10 levels in liver and VAT to address this point. It could also be interesting to look at possible differences between DNT and not DNT treated animals.

Answer: We thank the reviewer for this suggestion. As reviewer's suggestion, we detected CXCR3 ligand *CXCL9*, *CXCL10*, *CXCL11* mRNA levels in adipose tissues and liver tissues of HFD, MCD and CD-HFD-fed mice. As shown in Figure 20 in this response letter, *CXCL9* and *CXCL10* levels increased in adipose and liver tissues in NASH models, there were no significantly differences of these chemokines in mice with or without DNT treatment. These results were also shown in the revised manuscript (Figure S5C).

Figure 20. *CXCL9*, *CXCL10*, *CXCL11* mRNA levels in adipose and liver tissues of HFD, MCD and CD-HFD-fed mice.

2. Since more than half of the results refer to DNT and macrophage interaction, this topic should be mentioned in the title, e.g. “Double negative T-cells prevent the development and progression of nonalcoholic steatohepatitis by suppression of M1 macrophages”.

Answer: We thank the reviewer for this insightful suggestion. We changed the manuscript title as “Double negative T-cells prevent the development and progression of nonalcoholic steatohepatitis by suppression of Th17 cells and M1 macrophages”

3. The procedure of evaluation of the transcriptome sequencing study (Figure 5A), biological process (Figure 5B) and KEGG pathway analysis should be described in the methods section at least briefly.

Answer: Thank the reviewer’s suggestion, and we have added a section in the methods to describe the transcriptome sequencing study.

“Transcriptome sequencing analysis

The DNT incubation with CD3/CD28 antibodies were stimulated with IL-10 (50 ng/ml) for 24 hours, then total RNA was isolated from these DNT. Transcriptome sequencing libraries were generated using NEBNext® Ultra™ RNA Library Prep Kit for Illumina® (NEB, USA) following manufacturer’s recommendations and sequenced on an Illumina HiSeq platform (Illumina, San Diego, CA). Sequences were aligned to the reference genome with TopHat and processed with Cufflinks, which quantifies each transcript in each sample using reference annotations produced by the University of California Santa Cruz UCSC. Differentially expressed genes with a fold change of ≥ 1.5 and $\text{padj} < 0.05$ between with IL-10 treated and without IL-10 treated DNT were submitted to GO and KEGG enrichment analysis, which uses unbiased methods to assess pathway enrichment. The mRNA sequencing data described in this study were uploaded to the National Center for Biotechnology Information (NCBI) Gene Expression Omnibus (accession no. GSE134346).”

4. CD-HFD should be explained in the abbreviations sections.

Answer: We thank the reviewer's suggestion, and we have added CD-HFD as Choline-deficient High Fat Diet in the abbreviations section.

5. Bodyweight development should be shown for MCD- and CD-HFD-fed mice, for the last group especially before and after DNT transfer.

Answer: We thank the reviewer for this insightful suggestion. As shown in Figure 21 in this response letter, we have detected the MCD and CD-HFD-fed mice bodyweight before and after DNT transfer. In the MCD-fed NASH model, the C57BL/6 mice received an adoptive transfer of 5×10^6 CD45.1-positive DNT (purity > 97%) at 0 week and then were fed the MCD for 4 weeks. There are no significantly difference of MCD-fed mice bodyweight with or without DNT transfer at 4 weeks. In the CD-HFD-fed NASH model, C57BL/6 mice were fed a CD-HFD for 12 weeks, received a transfer of 5×10^6 CD45.1-positive DNT (purity > 97%), and then continued to be fed the CD-HFD for another 4 weeks. Compared with CD-HFD-fed mice without DNT treated, transferring of DNT could decrease the bodyweight of CD-HFD-fed mice at 16 weeks. These results were also shown in the revised manuscript (Figure S1A and S3A).

Figure 21. Mice bodyweight in MCD and CD-HFD-fed mice with or without DNT transferred.

6. Especially graphs for mRNA analysis could be bigger for a better legibility.

Answer: We thank the reviewer's advice and we have enlarged the graphs of mRNA analysis.

7. Picture size of histological slides should be enlarged for better perceptibility.

Answer: Thank the reviewer's advice. We re-select field of histological slides and take pictures with 100× magnification to enlarge the picture size to contain portal fields and central veins.

8. Regarding supplemental Figure 1F: Authors show Glucose levels after 12 weeks. Does this show the time point before DNT transfer? Please clarify in the description.

Answer: We thank the reviewer for this insightful comment. In the CD-HFD-fed NASH model, C57BL/6 mice were fed a CD-HFD for 12 weeks, and we transferred 5×10^6 CD45.1-positive DNT (purity > 97%) at this time point. After DNT transfer, the mice were continued fed with CD-HFD for another 4 weeks.

9. Regarding supplemental Figure 1: Please state the number of samples “n” for CD-HFD-fed illustrations.

Answer: We thank the reviewer for raising this important question. The NCD group, n=5; CD-HFD group, n=4; CD-HFD+DNT group, n=4. We have added this in the figure legends.

10. In several graphs the lines showing significant differences are not of the same strength. Authors should standardize their illustrations.

Answer: We thank the reviewer's advice and we have standardized the illustrations in revised manuscript.

11. On page 15 (discussion) the word “of” is missing in the first sentence.

Answer: We thank the reviewer's advice and changed the sentence as “The suppression of M1 macrophages and Th17 cells by DNT might be the major mechanism by which

DNT transfer prevent NASH development.”

12. IFN- γ is spelled as “IFN-r”, e.g. in Figure 2 E and supplemental Figure 1D. Please change this into correct spelling.

Answer: We thank the reviewer’s suggestion and have corrected it in revised manuscript.

13. In Figure 4C macrophages were co-cultured with DNT and TNF α secretion was examined. Please describe more precisely, if this is only examined from M1 macrophages. Moreover it would also be interesting to test for IL-6 and IFN- γ secretion.

Answer: We thank the reviewer’s kind advice. In Figure 4C, we cocultured CD45.1-positive bone marrow derived macrophages with DNT at different ratio for 24 hours *in vitro*. And then the macrophage TNF- α secretion was detected by flow cytometry (n = 5). As the reviewer suggested, we have also tested IL-6 and IFN- γ secretion in these CD45.1-positive macrophages. As shown in Figure 22 in this response letter, the DNT could significantly inhibit macrophages TNF- α , IFN- γ but not IL-6 secretion. These results were also shown in the revised manuscript (Figure 4C).

Figure 22. DNT inhibited macrophages TNF- α , IFN- γ secretion, but not IL-6 secretion.

References

1. Juvet, S.C. and L. Zhang, *Double negative regulatory T cells in transplantation and autoimmunity: recent progress and future directions*. Journal of Molecular Cell Biology, 2012. **4**(1): p. 48-58.
2. Hillhouse, E.E., J.S. Delisle, and S. Lesage, *Immunoregulatory CD4(-) CD8(-) T cells as a potential therapeutic tool for transplantation, autoimmunity, and cancer*. Frontiers in Immunology, 2013. **4**.
3. Zhang, D., et al., *New differentiation pathway for double-negative regulatory T cells that regulates the magnitude of immune responses*. Blood, 2007. **109**(9): p. 4071-4079.
4. Chen, W.H., et al., *Both infiltrating regulatory T cells and insufficient antigen presentation are involved in long-term cardiac xenograft survival*. Journal of Immunology, 2007. **179**(3): p. 1542-1548.
5. Zhang, D., et al., *Adoptive cell therapy using antigen-specific CD4(-)CD8(-) T regulatory cells to prevent autoimmune diabetes and promote islet allograft survival in NOD mice*. Diabetologia, 2011. **54**(8): p. 2082-2092.
6. Fischer, K., et al., *Isolation and characterization of human antigen-specific TCR alpha beta(+) CD4(-)CD8(-) double negative regulatory T cells*. Blood, 2005. **106**(11): p. 924a-

924a.

7. Zhang, Z.X., et al., *Double-negative T cells, activated by xenoantigen, lyse autologous B and T cells using a perforin/granzyme-dependent, fas-fas ligand-independent pathway*. Journal of Immunology, 2006. **177**(10): p. 6920-6929.
8. Li, W.X., et al., *Ex vivo converted double negative T cells suppress activated B cells*. International Immunopharmacology, 2014. **20**(1): p. 164-169.
9. Gao, J.F., et al., *Regulation of antigen-expressing dendritic cells by double negative regulatory T cells*. European Journal of Immunology, 2011. **41**(9): p. 2699-2708.
10. He, K.M., et al., *Donor double-negative Treg promote allogeneic mixed chimerism and tolerance*. European Journal of Immunology, 2007. **37**(12): p. 3455-3466.
11. Zhang, Z.X., et al., *Identification of a previously unknown antigen-specific regulatory T cell and its mechanism of suppression*. Nature Medicine, 2000. **6**(7): p. 782-789.
12. Young, K.J., et al., *Inhibition of graft-versus-host disease by double-negative regulatory T cells*. Journal of Immunology, 2003. **171**(1): p. 134-141.
13. McIver, Z., et al., *Double-negative regulatory T cells induce allotolerance when expanded after allogeneic haematopoietic stem cell transplantation*. British Journal of Haematology, 2008. **141**(2): p. 170-178.
14. Duncan, B., et al., *Double Negative (CD3(+)/4(-)8(-)) TCR alpha beta Splenic Cells from Young NOD Mice Provide Long-Lasting Protection against Type 1 Diabetes*. Plos One, 2010. **5**(7).
15. Crispin, J.C., et al., *Expanded Double Negative T Cells in Patients with Systemic Lupus Erythematosus Produce IL-17 and Infiltrate the Kidneys*. Journal of Immunology, 2008. **181**(12): p. 8761-8766.
16. Crispin, J.C. and G.C. Tsokos, *Human TCR-alpha beta(+) CD4(-) CD8(-) T Cells Can Derive from CD8(+) T Cells and Display an Inflammatory Effector Phenotype*. Journal of Immunology, 2009. **183**(7): p. 4675-4681.
17. Doreau, A., et al., *Interleukin 17 acts in synergy with B cell-activating factor to influence B cell biology and the pathophysiology of systemic lupus erythematosus*. Nature Immunology, 2009. **10**(7): p. 778-U142.
18. Alunno, A., et al., *CD4(-)CD8(-) T-cells in primary Sjogren's syndrome: Association with the extent of glandular involvement*. Journal of Autoimmunity, 2014. **51**: p. 38-43.
19. Alunno, A., et al., *IL-17-producing CD4(-)CD8(-) T cells are expanded in the peripheral blood, infiltrate salivary glands and are resistant to corticosteroids in patients with primary Sjogren's syndrome*. Annals of the Rheumatic Diseases, 2013. **72**(2): p. 286-292.
20. Zhao, X., et al., *A novel differentiation pathway from CD4(+) T cells to CD4(-) T cells for maintaining immune system homeostasis*. Cell Death & Disease, 2016. **7**.
21. Zhang, Z.X., K. Young, and L. Zhang, *CD3(+)/CD4(-)CD8(-) alpha beta-TCR+ T cell as immune regulatory cell*. Journal of Molecular Medicine-Jmm, 2001. **79**(8): p. 419-427.
22. Fischer, K., et al., *Isolation and characterization of human antigen-specific TCR alpha beta(+) CD4(-)CD8(-) double-negative regulatory T cells*. Blood, 2005. **105**(7): p. 2828-2835.
23. Crispin, J.C., et al., *Expanded double negative T cells in patients with systemic lupus erythematosus produce IL-17 and infiltrate the kidneys*. J Immunol, 2008. **181**(12): p. 8761-6.

24. Crispin, J.C. and G.C. Tsokos, *Human TCR-alpha beta+ CD4- CD8- T cells can derive from CD8+ T cells and display an inflammatory effector phenotype*. J Immunol, 2009. **183**(7): p. 4675-81.
25. Doreau, A., et al., *Interleukin 17 acts in synergy with B cell-activating factor to influence B cell biology and the pathophysiology of systemic lupus erythematosus*. Nat Immunol, 2009. **10**(7): p. 778-85.
26. Hillhouse, E.E., J.S. Delisle, and S. Lesage, *Immunoregulatory CD4(-)CD8(-) T cells as a potential therapeutic tool for transplantation, autoimmunity, and cancer*. Front Immunol, 2013. **4**: p. 6.
27. Juvet, S.C. and L. Zhang, *Double negative regulatory T cells in transplantation and autoimmunity: recent progress and future directions*. J Mol Cell Biol, 2012. **4**(1): p. 48-58.
28. Duncan, B., et al., *Double negative (CD3+ 4- 8-) TCR alphabeta splenic cells from young NOD mice provide long-lasting protection against type 1 diabetes*. PLoS One, 2010. **5**(7): p. e11427.
29. McIver, Z., et al., *Double-negative regulatory T cells induce allotolerance when expanded after allogeneic haematopoietic stem cell transplantation*. Br J Haematol, 2008. **141**(2): p. 170-8.
30. Zhang, D., et al., *Adoptive cell therapy using antigen-specific CD4⁺CD8⁻T regulatory cells to prevent autoimmune diabetes and promote islet allograft survival in NOD mice*. Diabetologia, 2011. **54**(8): p. 2082-92.
31. Liu, T., et al., *Combination of double negative T cells and anti-thymocyte serum reverses type 1 diabetes in NOD mice*. J Transl Med, 2016. **14**: p. 57.
32. Tian, D., et al., *Double negative T cells mediate Lag3-dependent antigen-specific protection in allergic asthma*. Nature Communications, 2019. **10**.
33. Lai, Z.W., et al., *Mechanistic Target of Rapamycin Activation Triggers IL-4 Production and Necrotic Death of Double-Negative T Cells in Patients with Systemic Lupus Erythematosus*. Journal of Immunology, 2013. **191**(5): p. 2236-2246.
34. Rodriguez-Rodriguez, N., et al., *Pro-inflammatory self-reactive T cells are found within murine TCR-alpha beta(+)CD4(-)CD8(-)PD-1(+) cells*. European Journal of Immunology, 2016. **46**(6): p. 1383-1391.
35. Larter, C.Z. and M.M. Yeh, *Animal models of NASH: Getting both pathology and metabolic context right*. Journal of Gastroenterology and Hepatology, 2008. **23**(11): p. 1635-1648.
36. Raubenheimer, P.J., M.J. Nyirenda, and B.R. Walker, *A choline-deficient diet exacerbates fatty liver but attenuates insulin resistance and glucose intolerance in mice fed a high-fat diet*. Diabetes, 2006. **55**(7): p. 2015-2020.
37. Fen, Z.G., et al., *DPP-4 Inhibition by Linagliptin Attenuates Obesity-Related Inflammation and Insulin Resistance by Regulating M1/M2 Macrophage Polarization*. Diabetes, 2016. **65**(10): p. 2966-2979.
38. Ono, Y., et al., *CD11c+M1-like macrophages (M Phi s) but not CD206+M2-like M Phi are involved in folliculogenesis in mice ovary*. Scientific Reports, 2018. **8**.
39. Cabalen, M.E., et al., *Chronic Trypanosoma cruzi infection potentiates adipose tissue macrophage polarization toward an anti-inflammatory M2 phenotype and contributes to diabetes progression in a diet-induced obesity model*. Oncotarget, 2016. **7**(12): p. 13400-13415.

40. Arimura, K., et al., *Crucial role of plasmacytoid dendritic cells in the development of acute colitis through the regulation of intestinal inflammation*. *Mucosal Immunology*, 2017. **10**(4): p. 957-970.
41. Hansen, L., et al., *E2-2 Dependent Plasmacytoid Dendritic Cells Control Autoimmune Diabetes*. *Plos One*, 2015. **10**(12).
42. Liao, X.F., et al., *Cutting Edge: Plasmacytoid Dendritic Cells in Late-Stage Lupus Mice Defective in Producing IFN- α* . *Journal of Immunology*, 2015. **195**(10): p. 4578-4582.
43. Eguchi, K. and R. Nagai, *Islet inflammation in type 2 diabetes and physiology*. *Journal of Clinical Investigation*, 2017. **127**(1): p. 14-23.
44. Zheng, M.J., J.L. Yu, and Z.G. Tian, *Characterization of the liver-draining lymph nodes in mice and their role in mounting regional immunity to HBV*. *Cellular & Molecular Immunology*, 2013. **10**(2): p. 143-150.
45. Racanelli, V. and B. Rehermann, *The liver as an immunological organ*. *Hepatology*, 2006. **43**(2): p. S54-S62.
46. Boor, P.P.C., et al., *Characterization of Antigen-Presenting Cell Subsets in Human Liver-Draining Lymph Nodes*. *Frontiers in Immunology*, 2019. **10**.
47. Barbier, L., et al., *Two lymph nodes draining the mouse liver are the preferential site of DC migration and T cell activation*. *Journal of Hepatology*, 2012. **57**(2): p. 352-358.

Reviewers' Comments:

Reviewer #1:

Remarks to the Author:

Review of NCOMMS-19-32808A-Z

Double negative T-cells prevent the development and progression of nonalcoholic steatohepatitis.

The revised paper is improved. However, significant concerns remain. Clearly, rapamycin reduces NAFLD, which is an inflammatory disease of the liver. Although the pro-inflammatory potential of DN T cells is addressed in the response to comments, these are largely omitted from the paper along with a large body of potentially supporting data embedded in the authors' response.

The findings of this paper remain surprising given that DN T cells are generally considered a pro-inflammatory subset within the T cell compartment and that rapamycin eliminates pro-inflammatory DN T cells via autophagy (1-5). Moreover, rapamycin also blocks pro-inflammatory mTOR activation within hepatocytes and abrogates the production of antiphospholipid antibodies in mice (6) and humans (4). The latter is a likely mechanism of action of rapamycin when blocking liver disease in SLE (7). The paper fails to address whether rapamycin blocks activation of mTOR within hepatocytes or it blocks the infiltration of the liver or pro-inflammatory function of DN T cells. These issues are critical for basic understanding and biological and clinical implications of this paper. A thorough discussion of alternative mechanisms of activation and function of DN T cells is lacking from this paper, some of which was included in the response to previous comments.

Specific comments:

Figure 8: this mechanistic diagram should include pro-inflammatory potential of DN T cells.

Figure S1: panel D shows a western blot. The loading control should be placed on the very same membrane which shows newly claimed changes in expression of specific genes. The GAPDH pattern appears different from SMA. Beta-actin and tubulin would be better controls. Moreover, GAPDH which is a metabolic enzyme linked to NAFLD in the past.

Figure S2: Effect of rapamycin on mTOR activation should be assessed in hepatocytes and compared to infiltrating DNT cells.

Figure S9: panel H shows a western blot. The loading control should be placed on the very same membrane which shows newly claimed changes in expression of specific genes. The GAPDH pattern appears different from either TNF-alpha or IL-10. Moreover, actin would be more appropriate than GAPDH which is a metabolic enzyme linked to NAFLD in the past.

Reference List

1. T. N. Caza et al., *Ann. Rheum. Dis.* 73, 1887 (2014).
2. Z.-W. Lai et al., *J. Immunol.* 191, 2236 (2013).
3. H. Kato, A. Perl, *J. Immunol.* 192, 4134 (2014).
4. Z. Lai et al., *Lancet* 391, 1186 (2018).
5. H. Kato, A. Perl, *Arthritis Rheumatol.* 70, 427 (2018).
6. Z. Oaks et al., *Arthritis Rheumatol.* 68, 2728 (2016).
7. Y. Liu et al., *Clin. Immunol.* 160, 319 (2015).

Reviewer #2:

Remarks to the Author:

The authors have satisfactorily addressed my concerns. The revised manuscript is much improved and the conclusions are now backed by data.

A minor issue appeared in the revised manuscript: In several figures, the authors show CXCL11, next to CXCL9 and 10. However, CXCL11 is non-functional in C57BL/6 mice and should thus be omitted from the figures, as this could be misleading.

Reviewer #3:

Remarks to the Author:

The authors greatly improved their study and underlined their findings by more evidence. Nearly all of my questions and suggestions were answered properly. Unfortunately scRNA-seq for macrophage differentiation was not performed. The authors cite older publications in their letter as evidence, that FACS analysis by itself is appropriate. Recent studies[1-3] underlie the new possibilities and more accurate differentiation by using scRNA-seq. Therefore I would still encourage them to use this method to subanalyse the macrophage population and polarisation in this study particularly as this study is facing immune cells as a treatment option.

[1] The Myeloid Cell Compartment-Cell by Cell.

Bassler K, Schulte-Schrepping J, Warnat-Herresthal S, Aschenbrenner AC, Schultze JL. *Annu Rev Immunol*. 2019 Apr 26;37:269-293. doi: 10.1146/annurev-immunol-042718-041728. Epub 2019 Jan 16. PMID: 30649988

[2] Single-cell RNA sequencing to explore immune cell heterogeneity. Papalexi E, Satija R. *Nat Rev Immunol*. 2018 Jan;18(1):35-45. doi: 10.1038/nri.2017.76. Epub 2017 Aug 7. PMID: 28787399

[3] Single-cell RNA sequencing reveals the heterogeneity of liver-resident immune cells in human. Zhao J, Zhang S, Liu Y, He X, Qu M, Xu G, Wang H, Huang M, Pan J, Liu Z, Li Z, Liu L, Zhang Z. *Cell Discov*. 2020 Apr 28;6:22. doi: 10.1038/s41421-020-0157-z. eCollection 2020. PMID: 32351704

Reviewer #1 (Remarks to the Author):

Review of NCOMMS-19-32808A-Z

Double negative T-cells prevent the development and progression of nonalcoholic steatohepatitis.

The revised paper is improved. However, significant concerns remain. Clearly, rapamycin reduces NAFLD, which is an inflammatory disease of the liver. Although the pro-inflammatory potential of DN T cells is addressed in the response to comments, these are largely omitted from the paper along with a large body of potentially supporting data embedded in the authors' response.

The findings of this paper remain surprising given that DN T cells are generally considered a pro-inflammatory subset within the T cell compartment and that rapamycin eliminates pro-inflammatory DN T cells via autophagy (1-5). Moreover, rapamycin also blocks pro-inflammatory mTOR activation within hepatocytes and abrogates the production of antiphospholipid antibodies in mice (6) and humans (4). The latter is a likely mechanism of action of rapamycin when blocking liver disease in SLE (7). The paper fails to address whether rapamycin blocks activation of mTOR within hepatocytes or it blocks the infiltration of the liver or pro-inflammatory function of DN T cells. These issues are critical for basic understanding and biological and clinical implications of this paper. A thorough discussion of alternative mechanisms of activation and function of DN T cells is lacking from this paper, some of which was included in the response to previous comments.

Answer: We thank the reviewer for raising these important questions.

1) DNT cells have been found to be both pro- and anti-inflammatory functions.

Although double negative T cells (DNT) only account for 1%-3% of total T lymphocytes in the peripheral blood and lymph organs of humans and mice, these rare T cells have been found to be important in various disease presentations.

DNT cells have been proposed to be essential in maintenance of immune homeostasis and self-tolerance. DNT highly express perforin, granzyme B, FasL and exhibit strong suppressive activity toward T cells (1-4), B cells (5, 6), dendritic cells (7), and NK cells (8), which could suppress the immune response and provide significant protection against allograft rejection, graft-versus-host disease (GVHD), autoimmune diseases, and ischemia-reperfusion injury (1, 3, 9-12).

On the other hand, as the reviewer mentioned, reports do suggest that DNT are involved in

systemic inflammation and tissue damage under autoimmune or inflammatory conditions. In patients with systemic lupus erythematosus (SLE) (13-15), primary Sjögren's syndrome (pSS) (16, 17), and psoriasis (18), DNT that expands in peripheral blood and inflamed tissues are the major producers of the proinflammatory cytokine IL-17 and are believed to be pathogenic in these autoimmune diseases.

According to previous studies, DNT can develop in thymus (19, 20) and also generate from CD8+ (21) or CD4+ T cells (1). Among these DNT, CD8+ T cell derived DNT could produce proinflammatory cytokines (IL-17) with obvious effector phenotype. On the other hand, DNT generated from CD4+ T cell are promising for maintenance of immune homeostasis and self-tolerance. The DNT we used in this study are converted from CD4+ T cells, to better distinguish these DNT from thymus developed or CD8 derived DNT, in this response letter and revised manuscript, we have changed the name of “DNT” to “CD4 T cell converted DNT (cDNT)”.

2) Some liver resident DNT could also secrete IL-17, the pro-inflammatory cytokines.

In order to further confirm the characterizations of DNT, we examined cytokines secreted from liver resident DNT in NCD- or MCD-fed mice. As shown in Figure 1, liver resident DNT could also secrete IL-17, a pro-inflammatory cytokine. Compared with control mice, IL-17 secreted from liver resident DNT increased significantly in MCD-fed NASH mice, which was consistent with the reviewer mentioned in SLE models (13). These results were also shown in the revised manuscript (Figure S2D-2E).

Figure 1. The IL-17 secretion of liver resident DNT in NCD- or MCD-fed mice with cDNT or rapamycin treatment.

3) Compared with CD4 T and liver resident DNT, cDNT we used in this study are defined as one type of regulatory T cells.

3.1 Functions of cDNT: Our previous studies demonstrated that CD4 T cells could convert to DNT *in vivo* and *in vitro* (1, 22). These cDNT could significantly suppress CD4+CD25- T cells and B cells immune response, mainly through the perforin/granzyme B pathway (6). The adoptive transfer of cDNT could prevent and reverse the onset of autoimmune diabetes and prolong islet graft survival while preserving antigen specificity (1, 3, 23). Recently, we also found cDNT have antigen-specific protection in allergic asthma (24).

3.2 Characterizations of cDNT: These cDNT do not express IL-2, IL-4 and IL-17 (1, 2, 22), are not like IL-17 producing pro-inflammatory DNT.

3.2.1. The cytokine secretion of cDNT cells *in vitro*.

In order to better understand the roles of these CD4 derived DNT, we firstly detected the activated CD4+ T cells and cDNT cytokines secretion and immunosuppressive molecule expression *in vitro*. As shown in Figure 2A-C, compared with activated CD4+ T cells, cDNT had extremely low or no secretion of IL-2, IL-4, IL-6, IL-10, IL-13, IL-17A, IL-21 and TGF- β , except for IFN- γ . However, compared with CD4+ T cells, cDNT highly expressed Granzyme B, the functional molecular of regulatory DNT.

3.2.2. The cytokine secretion of cDNT cells *in vivo*.

We also transferred cDNT into MCD-fed mice. Four weeks later, we detected the ability of cDNT to produce cytokines *in vivo*. As shown in Figure 2D, these cDNT also had extremely low or no secretion of IL-2, IL-4, IL-6, IL-10 and IL-17, including IFN- γ *in vivo*, but highly expressed Granzyme B. These observations indicated that cDNT are regulatory immune cells and are not proinflammatory cells. These results were also shown in the revised manuscript (Figure S6A-6D).

Figure 2. The cytokines of cDNT secreted *in vitro* and *in vivo*.

3.2.3. The characterizations of liver resident DNT and cDNT cells *in vivo*.

To better distinguish cDNT we used in this study with proinflammatory DNT, we also detected the cytokines secretion in transferred cDNT and liver resident DNT in MCD-fed mice. As shown in Figure 3, liver resident DNT could secrete IL-17, however, cDNT did not secrete IL-17. These results were also shown in the revised manuscript (Figure S2F-2G).

Figure 3. The cytokines of cDNT and liver resident DNT secreted in MCD-fed NASH liver *in vivo*.

4) The effects of rapamycin on liver resident DNT and cDNT.

As the reviewer suggested, we also examined the cytokines secretion of liver resident DNT and cDNT cells with or without rapamycin stimulation.

4.1 Rapamycin inhibited liver resident DNT proinflammation *in vivo*.

As shown in Figure 1 in this response letter, in MCD-fed NASH mice, rapamycin could inhibit liver resident DNT IL-17 secretion, while the transfer of cDNT did not influence liver resident DNT IL-17 secretion significantly.

4.2 Rapamycin might not be important for cDNT functions.

In this study, we mainly found IL-10 secreted by M2 macrophages decreases the survival and function of cDNT to protect M2 macrophages from cDNT-mediated lysis. In order to determine the roles of mTOR signal pathway in IL-10 stimulated cDNT, we also examined pi-mTOR expression of cDNT with or without IL-10 stimulation *in vitro*. As shown in Figure 4A and 4B in this response letter, IL-10 stimulation did not increase pi-mTOR expression, which indicated mTOR signal pathway might not important for cDNT during IL-10 stimulation. Meanwhile, as shown in Figure 4C, the cDNT had extremely low secretion of IL-2, IL-4, IL-6, IL-10 and IL-17, and rapamycin stimulation did not influence cDNT cytokine secretion. However, rapamycin stimulation could suppress IFN- γ secretion in cDNT (Figure 4C in response letter). These results were also shown in the revised manuscript (Figure S11).

Figure 4. The roles of mTOR pathway in cDNT cytokine secretion.

Meanwhile, we have added a section in the discussion to describe the mTOR signal pathway roles in cDNT activation and function as follows.

“The DNT play critical and diverse roles in the immune system, and they have both pro-inflammatory and anti-inflammatory functions. In patients with systemic lupus erythematosus (SLE) (13-15), primary Sjögren's syndrome (pSS) (16, 17), and psoriasis (18), DNT that expands in peripheral blood and inflamed tissues are the major producers of the proinflammatory cytokine IL-17 and are believed to be pathogenic in these autoimmune diseases. However, the CD4 converted DNT (cDNT) we used in this study, had extremely low or no secretion of pro-inflammatory cytokines, highly expressed Granzyme B, which were defined as one type of regulatory T cells. These cDNT could prevent or reverse the onset of type 1 diabetes, allergic asthma, liver injury, and induce allograft tolerance (1, 3, 22-24).”

“In SLE patients, mTORC1 activity is prominently increased in DNT, and play important roles in DNT pro-inflammation (13, 25). In this study, IL-10 stimulation did not increase pi-mTOR expression of cDNT, however, IL-10 upregulated PI3K-AKT and MAPK pathway

related gene expression, which indicated that these signaling pathways may be involved in the IL-10-mediated cDNT function. The mechanisms of IL-10 regulating cDNT activation and function need to be further studied.”

Specific comments:

Figure 8: this mechanistic diagram should include pro-inflammatory potential of DN T cells.

Answer: We thank the reviewer for raising these questions.

Indeed, the cDNT we used in this study did not secrete pro-inflammatory cytokines, such as IL-17, and these cells were defined as one type of regulatory T cells. To better distinguish the cDNT we used in this study with proinflammatory DNT, we replace “DNT” with “cDNT” and added the origin CD4+ T cells in the mechanistic diagram.

Figure S1: panel D shows a western blot. The loading control should be placed on the very same membrane which shows newly claimed changes in expression of specific genes. The GAPDH pattern appears different from SMA. Beta-actin and tubulin would be better controls. Moreover, GAPDH which is a metabolic enzyme linked to NAFLD in the past.

Answer: We thank the reviewer for the insightful comments.

We have added β -actin as the loading control. As shown in Figure 5, cDNT transfer significantly decreased α -SMA expression in MCD-fed mice. These results were also shown in the revised manuscript (Figure S1D).

Figure 5. α -SMA expression in liver tissues of NASH mice with or without cDNT transferred.

Figure S2: Effect of rapamycin on mTOR activation should be assessed in hepatocytes and compared to infiltrating DNT cells.

Answer: We thank the reviewer for this suggestion.

mTOR signal pathway plays important roles in hepatocytes lipid accumulation. Activation of the mTORC1 leads to the phosphorylation of the downstream effector of mTORC1: p70 S6 kinase (S6K1) and eIF4E-binding protein 1 (4E-BP1), which is an important for SREBP activation, lipogenesis, and hepatic steatosis (26). Many studies reported mTOR expression increased in hepatocytes of NASH mice and patients (27-29). Prolonged treatment with rapamycin sustained suppression of mTOR activity (30). Rapamycin, blocking mTORC signal, obviously ameliorated hepatic steatosis and liver injury through reducing SREBP-1c-dependent lipogenesis and promoting PPAR α -mediated fatty acid oxidation (31). Rapamycin treatment also enhanced hepatocytes autophagy and decreased hepatocytes lipid droplets accumulation, inflammation and ER stress (32-34).

Rapamycin might not important for cDNT functions. In this study, we mainly found IL-10 played important roles in regulating cDNT functions. However, IL-10 stimulation did not increase pi-mTOR expression, which indicated mTOR signal pathway might not important for cDNT, especially after IL-10 stimulation. Meanwhile, we also examined whether rapamycin, the inhibitor of mTOR signal pathway, blocked cDNT functions. As shown in Figure 4 in this response letter, the cDNT had extremely low secretion of IL-17, and rapamycin stimulation did not influence IL-17 secretion. These results were also shown in the revised manuscript (Figure S6A-D).

Meanwhile, in previous response letter and manuscript, as the reviewer suggested. we mainly used rapamycin as a positive control to confirm the roles of cDNT in ameliorating NASH development. The effects of rapamycin in hepatocytes and cDNT cells were not closely related to our research. We will further study the roles of rapamycin in regulating cDNT functions in the future.

Figure S9: panel H shows a western blot. The loading control should be placed on the very same membrane which shows newly claimed changes in expression of specific genes. The GAPDH pattern appears different from either TNF-alpha or IL-10. Moreover, actin would be more appropriate than GAPDH which is a metabolic enzyme linked to NAFLD in the past.

Answer: We thank the reviewer for the insightful comments.

As the reviewer suggestion, we have added β -actin as the loading control. As shown in Figure 6 in this response letter, cDNT transfer significantly decreased TNF- α expression in MCD-fed mice, while IL-10 expression did not change significantly after cDNT transfer. These results were also shown in the revised manuscript (Figure S9H).

Figure 6. TNF- α and IL-10 expression in liver tissues of NASH mice with or without cDNT transferred.

References:

Reviewer #2 (Remarks to the Author):

The authors have satisfactorily addressed my concerns. The revised manuscript is much improved and the conclusions are now backed by data.

A minor issue appeared in the revised manuscript: In several figures, the authors show CXCL11, next to CXCL9 and 10. However, CXCL11 is non-functional in C57BL/6 mice and should thus be omitted from the figures, as this could be misleading.

Answer: We thank the reviewer for this suggestion. As the reviewer suggested, we have omitted CXCL11 from the article and the figures.

Reviewer #3 (Remarks to the Author):

The authors greatly improved their study and underlined their findings by more evidence. Nearly all of my questions and suggestions were answered properly. Unfortunately scRNA-seq for macrophage differentiation was not performed. The authors cite older publications in their letter as evidence, that FACS analysis by itself is appropriate. Recent studies[1-3] underlie the new possibilities and more accurate differentiation by using scRNA-seq. Therefore I would still encourage them to use this method to subanalyse the macrophage population and polarisation in this study particularly as this study is facing immune cells as a treatment option.

Answer: We thank the reviewer for the insightful comments.

As the reviewer mentioned in the references, enormous plasticity and heterogeneity of macrophages play important roles for NASH development. And single-cell RNA sequencing is a better method to analyze the macrophage population and polarization.

We totally agree with the reviewer, scRNA-seq data would be extremely helpful to further reveal the mechanisms of DNT treatment in NASH. However, due to the space limit of the current manuscript (already contained 8 figures and 11 supplementary figures), it is hard to include a lot of new figures and tables generated by the tons of data from scRNA-seq. We believe that after the significant guidance of the reviewers, our current revised manuscript is greatly improved and more evidence are provided to support the hypothesis of this research. We will make a discussion about the weaknesses of our finding as the reviewers mentioned as “In this study, we mainly found cDNT selectively suppressed M1 macrophages, but not M2

macrophages during NASH development. Recent studies underlie the new possibilities and more accurate differentiation of hepatic macrophages by using scRNA-seq (35-37), thus scRNA-seq study might be more helpful to further reveal the macrophage population and polarization after cDNT transfer, we will perform the study in the future.”

Although we could not add scRNA-seq data in this manuscript, as the reviewer suggested, we have prepared to do a new scRNA-seq study in next 1-2 years. We have selected the hepatic macrophages (defined as CD45⁺Ly6G⁻CD11b^{high}F4/80^{int}) in NCD and MCD-fed mice through FACS, and then performed scRNA-seq.

Firstly, we confirmed the *Adgre1*, *Itgam*, *Ptprc* and *Ly6g* expression in these macrophages, which were consistent with F4/80, CD11b, CD45 and Ly6G expression in protein levels. As shown in Figure 7, UMAP analysis of liver macrophages showed 12 clusters. The proportions of clusters 0, 1, 4 were increased significantly in MCD-fed mice, while clusters 2, 3 decreased remarkably. Then, we will identify these clusters markers and functions, and confirm whether cDNT transfer could influence the clusters proportions and functions. We also plan to identify novel tissue macrophage subsets, and examine whether cDNT transfer could influence these cells proportions, functions, and explore the mechanisms in the future.

Figure 7. scRNA-seq analysis of hepatic macrophages in NCD- and MCD-fed mice.

Reference

1. Zhang D, Yang W, Degauque N, Tian Y, Mikita A, Zheng XX. New differentiation pathway for double-negative regulatory T cells that regulates the magnitude of immune responses. *Blood*. 2007;109(9):4071-9.
2. Chen WH, Diao J, Stepkowski SM, Zhang L. Both infiltrating regulatory T cells and insufficient antigen presentation are involved in long-term cardiac xenograft survival. *J Immunol*. 2007;179(3):1542-8.
3. Zhang D, Zhang W, Ng TW, Wang Y, Liu Q, Gorantla V, et al. Adoptive cell therapy using antigen-specific CD4(-)CD8(-) T regulatory cells to prevent autoimmune diabetes and promote islet allograft survival in NOD mice. *Diabetologia*. 2011;54(8):2082-92.
4. Fischer K, Voelkl S, Przybylski GK, Schmidt CA, Andreesen R, Mackensen A. Isolation and characterization of human antigen-specific TCR alpha beta(+) CD4(-)CD8(-) double negative regulatory T cells. *Blood*. 2005;106(11):924a-a.
5. Zhang ZX, Ma YX, Wang H, Arp J, Jiang JF, Huang XY, et al. Double-negative T cells, activated by xenoantigen, lyse autologous B and T cells using a perforin/granzyme-dependent, fas-fas ligand-independent pathway. *J Immunol*. 2006;177(10):6920-9.
6. Li WX, Tian Y, Li Z, Gao J, Shi W, Zhu JY, et al. Ex vivo converted double negative T cells suppress activated B cells. *International Immunopharmacology*. 2014;20(1):164-9.
7. Gao JF, McIntyre MSF, Juvet SC, Diao J, Li XJ, Vanama RB, et al. Regulation of antigen-expressing dendritic cells by double negative regulatory T cells. *Eur J Immunol*. 2011;41(9):2699-708.
8. He KM, Ma Y, Wang S, Min WP, Zhong R, Jevnikar A, et al. Donor double-negative Treg

- promote allogeneic mixed chimerism and tolerance. *Eur J Immunol.* 2007;37(12):3455-66.
9. Zhang ZX, Yang L, Young KJ, DuTemple B, Zhang L. Identification of a previously unknown antigen-specific regulatory T cell and its mechanism of suppression. *Nat Med.* 2000;6(7):782-9.
 10. Young KJ, DuTemple B, Phillips MJ, Zhang L. Inhibition of graft-versus-host disease by double-negative regulatory T cells. *J Immunol.* 2003;171(1):134-41.
 11. Mclver Z, Serio B, Dunbar A, O'Keefe CL, Powers J, Wlodarski M, et al. Double-negative regulatory T cells induce allotolerance when expanded after allogeneic haematopoietic stem cell transplantation. *Brit J Haematol.* 2008;141(2):170-8.
 12. Duncan B, Nazarov-Stoica C, Surls J, Kehl M, Bona C, Casares S, et al. Double Negative (CD3(+)/4(-)/8(-)) TCR alpha beta Splenic Cells from Young NOD Mice Provide Long-Lasting Protection against Type 1 Diabetes. *Plos One.* 2010;5(7).
 13. Crispin JC, Oukka M, Bayliss G, Cohen RA, Van Beek CA, Stillman IE, et al. Expanded Double Negative T Cells in Patients with Systemic Lupus Erythematosus Produce IL-17 and Infiltrate the Kidneys. *J Immunol.* 2008;181(12):8761-6.
 14. Crispin JC, Tsokos GC. Human TCR-alpha beta(+) CD4(-) CD8(-) T Cells Can Derive from CD8(+) T Cells and Display an Inflammatory Effector Phenotype. *J Immunol.* 2009;183(7):4675-81.
 15. Doreau A, Belot A, Bastid J, Riche B, Trescol-Biemont MC, Ranchin B, et al. Interleukin 17 acts in synergy with B cell-activating factor to influence B cell biology and the pathophysiology of systemic lupus erythematosus. *Nat Immunol.* 2009;10(7):778-U142.
 16. Alunno A, Carubbi F, Bistoni O, Caterbi S, Bartoloni E, Bigerna B, et al. CD4(-)CD8(-)

T-cells in primary Sjogren's syndrome: Association with the extent of glandular involvement. *J Autoimmun.* 2014;51:38-43.

17. Alunno A, Bistoni O, Bartoloni E, Caterbi S, Bigerna B, Tabarrini A, et al. IL-17-producing CD4(-)CD8(-) T cells are expanded in the peripheral blood, infiltrate salivary glands and are resistant to corticosteroids in patients with primary Sjogren's syndrome. *Ann Rheum Dis.* 2013;72(2):286-92.

18. Brandt D, Sergon M, Abraham S, Mabert K, Hedrich CM. TCR+ CD3(+) CD4(-) CD8(-) effector T cells in psoriasis. *Clin Immunol.* 2017;181:51-9.

19. Hayes SM, Li L, Love PE. TCR signal strength influences alphabeta/gammadelta lineage fate. *Immunity.* 2005;22(5):583-93.

20. Chapman JC, Chapman FM, Michael SD. The production of alpha/beta and gamma/delta double negative (DN) T-cells and their role in the maintenance of pregnancy. *Reprod Biol Endocrinol.* 2015;13:73.

21. Hedrich CM, Rauen T, Crispin JC, Koga T, Ioannidis C, Zajdel M, et al. cAMP-responsive element modulator alpha (CREMalpha) trans-represses the transmembrane glycoprotein CD8 and contributes to the generation of CD3+CD4-CD8- T cells in health and disease. *The Journal of biological chemistry.* 2013;288(44):31880-7.

22. Zhao X, Sun G, Sun X, Tian D, Liu K, Liu T, et al. A novel differentiation pathway from CD4(+) T cells to CD4(-) T cells for maintaining immune system homeostasis. *Cell Death Dis.* 2016;7.

23. Liu TH, Cong M, Sun GY, Wang P, Tian Y, Shi W, et al. Combination of double negative T cells and anti-thymocyte serum reverses type 1 diabetes in NOD mice. *J Transl Med.* 2016;14.

24. Tian D, Yang L, Wang S, Zhu YB, Shi W, Zhang CP, et al. Double negative T cells mediate Lag3-dependent antigen-specific protection in allergic asthma. *Nat Commun.* 2019;10.
25. Kato H, Perl A. Mechanistic Target of Rapamycin Complex 1 Expands Th17 and IL-4(+) CD4-CD8-Double-Negative T Cells and Contracts Regulatory T Cells in Systemic Lupus Erythematosus. *J Immunol.* 2014;192(9):4134-44.
26. Chen H. Nutrient mTORC1 signaling contributes to hepatic lipid metabolism in the pathogenesis of non-alcoholic fatty liver disease. *Liver Research.* 2020;4(1):15-22.
27. Wang Y, Shi M, Fu H, Xu H, Wei J, Wang T, et al. Mammalian target of the rapamycin pathway is involved in non-alcoholic fatty liver disease. *Mol Med Rep.* 2010;3(6):909-15.
28. Choi E, Kim W, Joo SK, Park S, Park JH, Kang YK, et al. Expression patterns of STAT3, ERK and estrogen-receptor alpha are associated with development and histologic severity of hepatic steatosis: a retrospective study. *Diagn Pathol.* 2018;13(1):23.
29. Kubrusly MS, Correa-Giannella ML, Bellodi-Privato M, de Sa SV, de Oliveira CPMS, Soares IC, et al. A role for mammalian target of rapamycin (mTOR) pathway in non alcoholic steatohepatitis related-cirrhosis. *Histol Histopathol.* 2010;25(9):1123-31.
30. Leontieva OV, Paszkiewicz GM, Blagosklonny MV. Comparison of rapamycin schedules in mice on high-fat diet. *Cell Cycle.* 2014;13(21):3350-6.
31. Zhao R, Zhu M, Zhou S, Feng W, Chen H. Rapamycin-Loaded mPEG-PLGA Nanoparticles Ameliorate Hepatic Steatosis and Liver Injury in Non-alcoholic Fatty Liver Disease. *Front Chem.* 2020;8:407.
32. Chen R, Wang Q, Song S, Liu F, He B, Gao X. Protective role of autophagy in

methionine-choline deficient diet-induced advanced nonalcoholic steatohepatitis in mice. *Eur J Pharmacol.* 2016;770:126-33.

33. Liu C, Liu L, Zhu HD, Sheng JQ, Wu XL, He XX, et al. Celecoxib alleviates nonalcoholic fatty liver disease by restoring autophagic flux. *Sci Rep.* 2018;8(1):4108.

34. Tanaka S, Hikita H, Tatsumi T, Sakamori R, Nozaki Y, Sakane S, et al. Rubicon inhibits autophagy and accelerates hepatocyte apoptosis and lipid accumulation in nonalcoholic fatty liver disease in mice. *Hepatology.* 2016;64(6):1994-2014.

35. Papalexi E, Satija R. Single-cell RNA sequencing to explore immune cell heterogeneity. *Nat Rev Immunol.* 2018;18(1):35-45.

36. Bassler K, Schulte-Schrepping J, Warnat-Herresthal S, Aschenbrenner AC, Schultze JL. The Myeloid Cell Compartment-Cell by Cell. *Annu Rev Immunol.* 2019;37:269-93.

37. Zhao J, Zhang S, Liu Y, He X, Qu M, Xu G, et al. Single-cell RNA sequencing reveals the heterogeneity of liver-resident immune cells in human. *Cell Discovery.* 2020;6(1).

Reviewers' Comments:

Reviewer #1:

Remarks to the Author:

The paper has been expanded with additional data that provide a shifting characterization of DN T cells that appear to protect from nonalcoholic steatohepatitis. Many of the changes are described in the response to comments, however, they have not been all represented in the paper itself. The newly offered designation of cDNT cells does not reflect their purported origin from CD4 T cells. The abbreviation of CD4-DNT cells would be more revealing. Such changes should also be reflected in the title which should read as follows:

"CD4-derived double negative T cells prevent the development and progression of nonalcoholic steatohepatitis by suppression of Th17 cells and M1 macrophages".

The role of mTOR and its discussion with respect to hepatitis is still lacking clarity. The authors should consider that their CD4-derived DN T cells may act as Tregs, which are in fact expanded in SLE by mTOR blockade both in vitro, and most importantly in vivo (Lancet, 391:1186-1196). Given the authors new data and shifting interpretation, mTOR blockade may have expanded Tregs that appear to have lost their CD4 expression. This potential mechanism of action needs to be addressed in the discussion. Along these lines, FoxP3+ Tregs need to be included among cells depicted in Figure 8. It is possible that the CD4-derived DN T cells may potentially act as Tregs.

Reviewer #1 (Remarks to the Author):

The paper has been expanded with additional data that provide a shifting characterization of DN T cells that appear to protect from nonalcoholic steatohepatitis. Many of the changes are described in the response to comments, however, they have not been all represented in the paper itself. The newly offered designation of cDNT cells does not reflect their purported origin from CD4 T cells. The abbreviation of CD4-DNT cells would be more revealing. Such changes should also be reflected in the title which should read as follows:

“CD4-derived double negative T cells prevent the development and progression of nonalcoholic steatohepatitis by suppression of Th17 cells and M1 macrophages”.

Answer: We thank the reviewer for this suggestion.

According to the reviewer’s suggestion and the word limited of the title, we have changed the title as follows:

“CD4 derived double negative T cells prevent the development and progression of nonalcoholic steatohepatitis.”

Meanwhile, as the reviewer suggested, CD4-DNT does reflect the origin of DNT, however, the term “CD4-DNT” might also make the readers confused with “CD4 and DNT” or “CD4 vs DNT”. We have used cDNT to represent CD4 derived double negative T cells in our previous published article¹. To be consistent with our previous report, we still use cDNT in revised manuscript and we hope the reviewer could agree with us that cDNT is acceptable.

To further clarify that these DNT are derived from CD4 T cells, we have added “cDNT” in Abbreviations and have also clearly given its full description as “CD4 converted DNT” in manuscript where cDNT first appears.

1. Sun G, et al. Critical role of OX40 in the expansion and survival of CD4 T-cell-derived double-negative T cells. *Cell Death Dis* 9, 616 (2018).

The role of mTOR and its discussion with respect to hepatitis is still lacking clarity. The authors should consider that their CD4-derived DN T cells may act as Tregs, which are in fact

expanded in SLE by mTOR blockade both in vitro, and most importantly in vivo (Lancet, 391:1186-1196). Given the authors new data and shifting interpretation, mTOR blockade may have expanded Tregs that appear to have lost their CD4 expression. This potential mechanism of action needs to be addressed in the discussion. Along these lines, FoxP3+ Tregs need to be included among cells depicted in Figure 8. It is possible that the CD4-derived DN T cells may potentially act as Tregs.

Answer: We thank the reviewer for the insightful comments.

1. In order to further clarify the role of mTOR in hepatitis, we added the following sentences in discussion.

“Activation of mTOR signal pathway plays important roles in liver diseases, and mTOR-inhibitor rapamycin could ameliorate liver fibrosis, autoimmune hepatitis through regulation Th17/Treg cell balance^{2, 3, 4}. The hepatic Foxp3+ Tregs in rapamycin-treated mice had significantly higher frequency and suppressive effects. Activated Treg with rapamycin treatment preferentially phosphorylated STAT5 and STAT3 and did not utilize the PI3K/mTOR pathway^{5, 6}.”

2. Actually, in previous study, we have compared the differences between cDNT and Treg cells^{7, 8}. Compared with Treg cells, the cDNT do not express Foxp3. On the contrary, cDNT express high levels of cytotoxic lymphocyte-related genes perforin and granzyme B. Thus, the cDNT are distinctive from Treg cells.

To further clearly address the reviewer’s concern, we also evaluated the effect of rapamycin, mTOR signal inhibitor, on CD4 expression in Foxp3+ Treg cells. As shown in Fig 1, rapamycin did not lower CD4 expression in Foxp3+ Treg cells in our study.

In addition, our study is focused on potential therapeutic effects of cDNT on NASH and NAFLD, is not a rapamycin related research.

Taken together, we do not think there are any data supporting that FoxP3+ Tregs need to be included among cells depicted in Figure 8. And we have added in Figure 8 that the cDNT cells is Granzyme B positive and Foxp3, IL-17 negative.

However, as the reviewer suggested, we still added a discussion in the paper as follows to make it more clear:

“However, the cDNT we used in this study, had low or no secretion of IL-17, highly expressed Granzyme B, which were defined as one type of regulatory T cells, **although no Foxp3 expression.**”

Figure 1. Rapamycin did not influence CD4 expression in Foxp3+ Treg cells

References:

1. Sun G, *et al.* Critical role of OX40 in the expansion and survival of CD4 T-cell-derived double-negative T cells. *Cell Death Dis* **9**, 616 (2018).
2. Holder BS, *et al.* Retinoic acid stabilizes antigen-specific regulatory T-cell function in autoimmune hepatitis type 2. *Journal of Autoimmunity* **53**, 26-32 (2014).
3. Gu L, Deng WS, Sun XF, Zhou H, Xu Q. Rapamycin ameliorates CCl4-induced liver fibrosis in mice through reciprocal regulation of the Th17/Treg cell balance. *Molecular Medicine Reports* **14**, 1153-1161 (2016).

4. Oo YH, Sakaguchi S. Regulatory T-cell directed therapies in liver diseases. *J Hepatol* **59**, 1127-1134 (2013).
5. Strauss L, Czystowska M, Szajnik M, Mandapathil M, Whiteside TL. Differential Responses of Human Regulatory T Cells (Treg) and Effector T Cells to Rapamycin. *Plos One* **4**, (2009).
6. McMahon G, Weir MR, Li XC, Mandelbrot DA. The Evolving Role of mTOR Inhibition in Transplantation Tolerance. *J Am Soc Nephrol* **22**, 408-415 (2011).
7. Zhang D, Yang W, Degauque N, Tian Y, Mikita A, Zheng XX. New differentiation pathway for double-negative regulatory T cells that regulates the magnitude of immune responses. *Blood* **109**, 4071-4079 (2007).
8. Zhao X, *et al.* A novel differentiation pathway from CD4(+) T cells to CD4(-) T cells for maintaining immune system homeostasis. *Cell Death Dis* **7**, e2193 (2016).